# Aberrant cortical development is driven by impaired cell cycle and translational control in a *DDX3X* syndrome model

**Mariah L Hoye[1], Lorenzo Calviello[2,3], Abigail J Poff[1], Nna-Emeka Ejimogu[1], Carly R Newman[1], Maya D Montgomery[1], Jianhong Ou[4,5], Stephen N Floor[6,7], Debra L Silver[1,4,5,8,9]\***

[1]Department of Molecular Genetics and Microbiology, Duke University Medical Center, Durham, United States; [2]Centre for Functional Genomics, Human Technopole, Milan, Italy; [3]Centre for Computational Biology, Human Technopole, Milan, Italy; [4]Department of Cell Biology, Duke University Medical Center, Durham, United States; [5]Duke Regeneration Center, Duke University Medical Center, Durham, United States; [6]Department of Cell and Tissue Biology, UCSF, San Francisco, United States; [7]Helen Diller Family Comprehensive Cancer Center, San Francisco, United States; [8]Department of Neurobiology, Duke University Medical Center, Durham, United States; [9]Duke Institute for Brain Sciences, Duke University Medical Center, Durham, United States

**\*For correspondence:**
debra.silver@duke.edu

**Abstract** Mutations in the RNA helicase, *DDX3X*, are a leading cause of Intellectual Disability and present as *DDX3X* syndrome, a neurodevelopmental disorder associated with cortical malformations and autism. Yet, the cellular and molecular mechanisms by which DDX3X controls cortical development are largely unknown. Here, using a mouse model of *Ddx3x* loss-of-function we demonstrate that DDX3X directs translational and cell cycle control of neural progenitors, which underlies precise corticogenesis. First, we show brain development is sensitive to *Ddx3x* dosage; complete *Ddx3x* loss from neural progenitors causes microcephaly in females, whereas hemizygous males and heterozygous females show reduced neurogenesis without marked microcephaly. In addition, *Ddx3x* loss is sexually dimorphic, as its paralog, *Ddx3y*, compensates for *Ddx3x* in the developing male neocortex. Using live imaging of progenitors, we show that DDX3X promotes neuronal generation by regulating both cell cycle duration and neurogenic divisions. Finally, we use ribosome profiling *in vivo* to discover the repertoire of translated transcripts in neural progenitors, including those which are DDX3X-dependent and essential for neurogenesis. Our study reveals invaluable new insights into the etiology of *DDX3X* syndrome, implicating dysregulated progenitor cell cycle dynamics and translation as pathogenic mechanisms.

## Editor's evaluation

The paper beautifully documents the cortical developmental defects associated with DDX3X loss, detailing both the morphological, transcriptional, cell cycle, and protein translational defects in visually striking detail.

## Introduction

The cerebral cortex is critical for higher order cognitive, motor, and sensory functioning. These processes rely upon embryonic development, when the proper number and types of neurons are

**eLife digest** During development, a complex network of genes ensures that the brain develops in the right way. In particular, they control how special 'progenitor' cells multiply and mature to form neurons during a process known as neurogenesis. Genetic mutations that interfere with neurogenesis can lead to disability and defects such as microcephaly, where children are born with abnormally small brains.

*DDX3X* syndrome is a recently identified condition characterised by intellectual disability, delayed acquisition of movement and language skills, low muscle tone and, frequently, a diagnosis of autism spectrum disorder. It emerges when certain mutations are present in the *DDX3X* gene, which helps to control the process by which proteins are built in a cell (also known as translation). The syndrome affects girls more often than boys, potentially because *DDX3X* is carried on the X chromosome. Many of the disease-causing mutations in the *DDX3X* gene also reduce the levels of DDX3X protein. However, exactly what genes *DDX3X* controls and how its loss impairs brain development remain poorly understood. To address this problem, Hoye et al. set out to investigate the role of *Ddx3x* in mice neurogenesis.

Experiments with genetically altered mice confirmed that complete loss of the gene indeed caused severe reduction in brain size at birth; just as in humans with mild microcephaly, this was only present in affected females. Further genetic studies revealed the reason for this: the closely related *Ddx3y* gene, which is only present on the Y (male) chromosome, helped to compensate for the loss of *Ddx3x* in the male mice.

Next, the effect of the loss of just one copy of *Ddx3x* on neurogenesis was examined by following how progenitor cells developed. This likely reflects DDX3X levels in patients with the syndrome. Loss of the gene made the cells divide more slowly and produce fewer mature nerve cells, suggesting that smaller brain size and brain malformations caused by mutations in *DDX3X* could be due to impaired neurogenesis. Finally, a set of further biochemical and genetic experiments revealed a key set of genes that are under the control of the DDX3X protein.

These results shed new light on how a molecular actor which helps to control translation is a key part of normal brain development. This understanding could one day help improve clinical management or treatments for *DDX3X* syndrome and related neurological disorders.

generated (*Kriegstein and Alvarez-Buylla, 2009*; *Lodato and Arlotta, 2015*; *Silver et al., 2019*). At the onset of cortical development, neuroepithelial cells divide symmetrically to expand the progenitor pool before transitioning into radial glial cells (RGCs). In the ventricular zone (VZ), RGCs undergo symmetric self-renewing divisions or asymmetric divisions to produce either intermediate progenitors (IPs) or neurons. IPs, also referred to as basal progenitors, divide 1–2 times in the sub-ventricular zone (SVZ) before terminally differentiating into neurons. While mouse neurogenesis largely relies on RGCs and IPs, humans have a more expansive neurogenic basal progenitor population. Excitatory neurons are produced in an inside-out fashion; wherein deep layers (V/VI) are the earliest born, followed by production of superficial neurons (IV-II/III). Newborn neurons migrate to the cortical plate (CP) where they differentially project axons depending on their laminar position. Impairments in these steps of cortical development can cause neurodevelopmental disorders, including microcephaly, intellectual disability (ID), and autism spectrum disorder (ASD) (*Polioudakis et al., 2019*; *Willsey et al., 2013*; *Willsey et al., 2021*).

*De novo* mutations in the RNA helicase, *DDX3X*, are one of the leading causes of ID in females and underlie *DDX3X* syndrome (*Beal et al., 2019*; *Lennox et al., 2020*; *Scala et al., 2019*; *Snijders Blok et al., 2015*; *Wang et al., 2018*). *DDX3X* is X-linked, which likely explains the preponderance of female cases; although an increasing number of *de novo* and inherited *DDX3X* mutations are found in males (*Kellaris et al., 2018*; *Nicola et al., 2019*). While *DDX3X* syndrome is characterized by ID, these individuals also commonly present with muscle tone and gait abnormalities, language deficits, abnormal brain MRIs (particularly white matter loss and corpus callosum defects), and many are diagnosed with ASD (*Johnson-Kerner et al., 1993*; *Lennox et al., 2020*; *Tang et al., 2021*). Indeed, *DDX3X* mutations have been identified in ASD cohorts, and *DDX3X* is considered a high-confidence Autism gene (*Iossifov et al., 2014*; *Ruzzo et al., 2019*; *Takata et al., 2018*; *Yuen, 2017*). *DDX3X*

mutations have also been linked to cancer progression, including medulloblastoma (*Jones, 2012*; *Pugh, 2012*; *Robinson et al., 2012*), many of which overlap with ID-associated mutations. This oncogenic role is consistent with conserved functions of DDX3X in cell cycle progression (*Chen et al., 2016a*; *Kotov et al., 2016*; *Li et al., 2014*; *Zhang and Li, 2021*). Notably, cell cycle duration is also strongly implicated in cortical development (*Arai et al., 2011*; *Lange et al., 2009*; *Pilaz et al., 2016*; *Pilaz et al., 2009*). Altogether, these clinical findings argue that *DDX3X* mutations are deleterious and suggest that cell cycle defects may underlie their association with cancer and ID.

Remarkably, over 100 *de novo* mutations have been identified in *DDX3X* syndrome, equally composed of nonsense/frameshift and missense (*Johnson-Kerner et al., 1993*). The former class likely results in *DDX3X* haploinsufficiency and/or hypomorphic loss-of-function (LoF) (*Figure 1A*). Consistent with this, in female mice, *Ddx3x* germline haploinsufficiency impairs postnatal brain architecture and causes behavioral deficits phenocopying aspects of human *DDX3X* syndrome (*Boitnott et al., 2021*). In addition, using transient CRISPR approaches we previously showed that acute *Ddx3x* depletion in a subset of cells perturbed progenitor and neuron number (*Lennox et al., 2020*). While these studies highlight the requirement of *Ddx3x* for cortical development, how it controls neurogenesis at the cellular and molecular level is unclear (*Figure 1B*). Indeed, the temporal and spatial requirements for DDX3X during cortical development are unknown, as are the dosage and sex-specific requirements.

DDX3X is a cytoplasmic RNA helicase that promotes translation of mRNAs with highly structured 5′ UTRs (*Calviello et al., 2021*; *Oh et al., 2016*). Thus, an intriguing possibility is that DDX3X controls cortical development by regulating translation. Although DDX3X translational targets have been characterized in immortalized cells (*Calviello et al., 2021*; *Oh et al., 2016*), they have not been identified in the developing brain *in vivo*. This hinders an understanding of how DDX3X molecularly controls brain development. Further, while translational control is essential for corticogenesis (*Blair et al., 2017*; *Hoye and Silver, 2021*; *Kraushar et al., 2014*; *Yang et al., 2014*; *Zahr et al., 2018*), there is limited genome-wide assessment of translation in the developing cortex, which restricts our understanding of this important layer of post-transcriptional regulation.

In this study, we use mouse genetics, live imaging of neural progenitors, and ribosome profiling to discover new underlying cellular and molecular mechanisms by which *Ddx3x* controls cortical development. Our study further reveals essential roles for translational regulation in directing cell fate decisions of the developing brain.

## Results

### Conditional knockout of *Ddx3x* in neural progenitors causes microcephaly and profound apoptosis

To understand how *Ddx3x* LoF impairs cortical development *in vivo*, we employed a previously generated floxed *Ddx3x* mouse (*Chen et al., 2016a*) and crossed it to *Emx1*-Cre (*Gorski et al., 2002*). This strategy removes *Ddx3x* from neural progenitors beginning at E9.5, as well as their progeny. As *Ddx3x* is X-linked, we generated *Emx1*-Cre conditional knockout (cKO) females (*Ddx3x^{lox/lox}*) and males (*Ddx3x^{lox/Y}*) and conditional heterozygous (cHet) females (*Ddx3x^{lox/+}*). To verify *Ddx3x* mRNA levels were reduced, we performed single molecule inexpensive fluorescence *in situ* hybridization (smiFISH) (*Tsanov et al., 2016*) at E12.5 (*Figure 1C*). There were less *Ddx3x* mRNA puncta in cHet females than in controls and significantly less *Ddx3x* mRNA puncta in cKO males and females (*Figure 1D*), demonstrating that *Ddx3x* levels are reduced.

We then quantitatively assessed *Ddx3x* levels in male and female embryonic brains. Towards this, we used the Cre reporter, *Rosa26^{Ai14}*, along with FACS to isolate TdTomato + cells from embryos, followed by RT-qPCR for *Ddx3x*. At E11.5, *Ddx3x* mRNA levels were ~30% reduced in cHet females,~40% reduced in cKO males and ~70% reduced in cKO females relative to their respective sex-matched controls (*Figure 1E*). At E14.5, we quantified a similar degree of reduction of *Ddx3x* in mutant brains (*Figure 1F*). The incomplete reduction of *Ddx3x* in females could reflect inefficiencies in recombination or preferential loss of mutant cells. Notably, control females expressed ~25% higher levels of *Ddx3x* than control males (*Figure 1G*), which was similarly evident in the quantification of smFISH data. We also quantified DDX3X protein levels using western blot analysis of E14.5 cortices (*Figure 1—figure supplement 1A*). DDX3X protein levels were higher in control females compared to males, similar to that seen at the RNA level. Further, compared to controls, cHet females showed decreased DDX3X

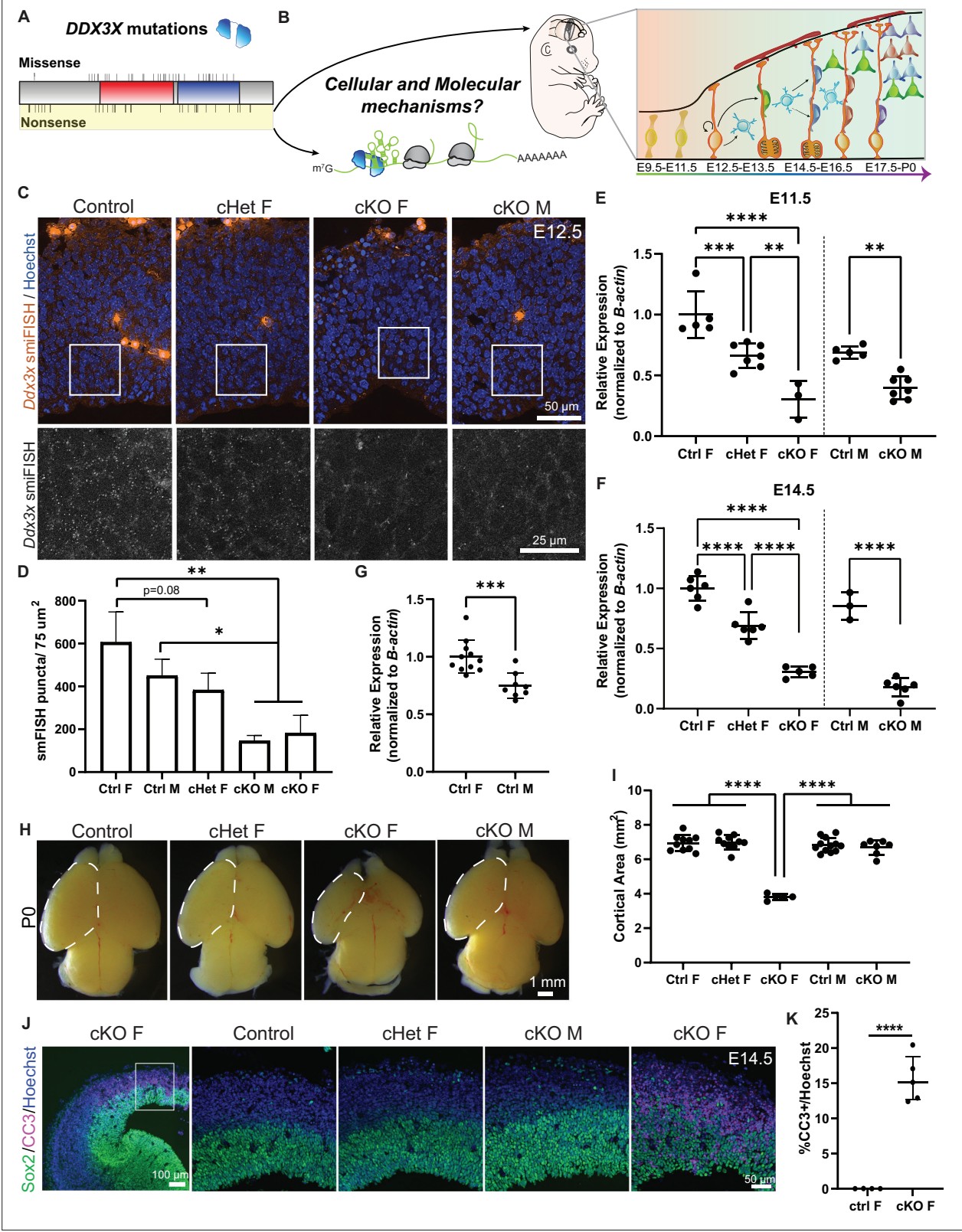

**Figure 1.** Conditional knockout of *Ddx3x* in neural progenitors using *Emx1*-Cre leads to microcephaly in female mice. (**A**) Schematic of DDX3X protein with human missense and nonsense mutations noted, along with helicase/RNA binding domains (red, blue). Nonsense mutations, highlighted in yellow, are predicted to act in a LoF manner. (**B**) (Left) DDX3X protein bound to an mRNA undergoing translation. (Right) Mouse embryo and corticogenesis showing neuroepithelial cells (light green), radial glial cells (RGCs, orange), intermediate progenitors (IPs, light blue), and neurons

*Figure 1 continued on next page*

*Figure 1 continued*

(multi-colored); *Figure 1B* adapted from Figure 1A and B from *Hoye and Silver, 2021*. This study asks how does *Ddx3x* LoF impair mouse embryonic cortical development at a cellular and molecular level? (**C**) Representative sections of smFISH for *Ddx3x* in control, cHet female, and cKO male and female E12.5 cortices. (**D**) Quantification of *Ddx3x* smFISH signal in respective genotypes at E12.5. n=2–3 embryos/condition (**E, F**) Validation of *Ddx3x* mRNA knockdown in Tdtomato + cells from female (**F**) (control, cHet, cKO) and male (**M**) (control, cKO) brains sorted via FACS at E11.5 (**E**) and E14.5 (**F**). n=3–7 embryos/condition. (**G**) Quantification of *Ddx3x* levels in Tdtomato + cells from control female and male brains. n=8–10 embryos/condition. (**H**) Representative whole mount images of control, cHet female, and cKO male and female brains at P0. (**I**) Quantification of cortical area at P0. n=5–12 embryos/condition. (**J**) Representative sections of E14.5 brains stained with Sox2 (green), CC3 (magenta) and Hoechst (blue) showing low-magnification on left panel, and high magnification on 4 panels to the right. (**K**) Quantification of CC3 + cells in E14.5 control and cKO female cortices. n=4–5 embryos/condition. Scale bars, indicated. Error bars, S.D. *p<0.05, **p<0.01, ***p<0.001, ****p<0.0001. One-way ANOVA with Tukey's (**D, E, F, I**), Student's unpaired, two-tailed t-test (**G, K**).

The online version of this article includes the following source data and figure supplement(s) for figure 1:

**Figure supplement 1.** *Ddx3x* loss from neural progenitors, but not neurons leads to microcephaly and apoptosis.

**Figure supplement 1—source data 1.** Western blot anlaysis of DDX3X in embryonic brain samples.

expression, and cKO males showed a striking reduction. These results further validate the *Ddx3x* cKO model, and are consistent with previous validation of this mouse (*Chen et al., 2016a*).

The significant difference in *Ddx3x* levels between control females and males is consistent with previous findings in mice and humans (*Tukiainen, 2017*; *Wu et al., 2014*) and suggests *Ddx3x* partially escapes X chromosome inactivation (XCI) in the embryonic cortex. Altogether, the qPCR, smFISH and western analyses demonstrate divergent *Ddx3x* levels in males and females in the developing brain. Further, these results validate and establish *Emx1*-Cre; *Ddx3x* conditional mice as a model to interrogate dose-dependent and sex-specific requirements of *Ddx3x* throughout cortical neurogenesis.

We next assessed gross cortical anatomy and size in *Ddx3x* mutant embryos. Strikingly, homozygous loss of *Ddx3x* from neural progenitors and their progeny led to profound microcephaly in cKO females at postnatal day 0 (P0) (*Figure 1H, I*). The decrease in cortical area began at E14.5, and by P0 was reduced by 50% (*Figure 1—figure supplement 1B*). In contrast, no significant reduction in cortical area was observed in cHet females or cKO males (*Figure 1H, I*). These data demonstrate that the developing brain requires *Ddx3x*.

As microcephaly is often associated with massive cell death, we assessed apoptosis in *Ddx3x* cKO females by immunostaining E12.5 and E14.5 sections for the apoptotic marker, cleaved caspase-3 (CC3). We observed ~5% of cells in cKO females were CC3+ at E12.5 and ~15% were CC3+ at E14.5 (*Figure 1J and K*; *Figure 1—figure supplement 1C, D*). Apoptotic cells were both Tuj1+ and Tuj1-, suggesting complete *Ddx3x* loss causes death of both progenitors and newborn neurons (*Figure 1—figure supplement 1E*). In contrast, and consistent with the absence of microcephaly in cHet females and cKO males, CC3 + cells were not detected at either E12.5 or E14.5 in these genotypes (*Figure 1J*; *Figure 1—figure supplement 1D*). Altogether, these data suggest widespread apoptosis is likely a substantial contributor to the reduced cortical area in cKO female mice.

The extensive apoptosis in cKO females could be due to *Ddx3x* requirements in neural precursors, or alternatively independent requirements in newborn neurons. To understand how *Ddx3x* loss impacts neuronal survival, we used *Neurod6*-Cre (herein referred to by alias, *Nex*-Cre), which is active in post-mitotic excitatory neurons beginning at E12.5, to generate *Nex*-Cre, cKO females (*Nex*-Cre;*Ddx3x^lox/lox^*) (*Goebbels et al., 2006*). *Ddx3x* depletion was validated using smFISH, which showed specific reduction in the cortical plate where neurons reside (*Figure 1—figure supplement 1F*). Unlike *Emx1*-Cre cKO females, we did not observe microcephaly or apoptosis in *Nex*-Cre, cKO females across cortical development (E13.5, E15.5, E17.5 (not shown) and P0; *Figure 1—figure supplement 1G-I*). This suggests that apoptosis of newborn neurons in the *Emx1*-Cre model is due to defects in neural precursors. Taken together, these data reinforce the significant role of *Ddx3x* in neural progenitors.

## *Ddx3x* loss is sexually dimorphic and DDX3Y can compensate for loss of DDX3X

While the vast majority of individuals with *DDX3X* syndrome are females, males can harbor either maternally inherited or *de novo* *DDX3X* mutations (*Kellaris et al., 2018*; *Nicola et al., 2019*). Thus, as males only have a single copy of *Ddx3x*, we predicted that loss of *Ddx3x* in males would

phenocopy cKO females. Surprisingly, cKO males did not display profound microcephaly or apoptosis (*Figure 1H–K*). Notably, males have a *Ddx3x* paralog, *Ddx3y*, on the Y chromosome which can compensate for *Ddx3x* at the translational level in immortalized cells (*Venkataramanan et al., 2021*). Thus, we postulated that *Ddx3y* may offset the loss of *Ddx3x* in the neocortex. We first examined if *Ddx3y* levels were altered following *Ddx3x* reduction. In E11.5 *Ddx3x* cKO male cortices, *Ddx3y* mRNA levels were significantly elevated 1.4 fold on average (*Figure 2A*). This suggests there is a transcriptional adaptation of *Ddx3y* in response to reduced *Ddx3x* in the embryonic cortex, consistent with findings in *Ddx3x* cKO hindbrains (*Patmore et al., 2020*).

We next probed the functional redundancy of DDX3X and DDX3Y by investigating requirements for *Ddx3y* during cortical development. To this end, we performed *in utero* electroporation (IUE) of E14.5 brains with *Ddx3y* sgRNA + Cas9 and pCAG-GFP to deplete *Ddx3y* in males (*Figure 2B*). Using FACS to isolate GFP+ cells from E17.5 brains, we quantified an average 53% reduction in *Ddx3y* mRNA levels following *Ddx3y* CRISPR-based depletion; there was no effect upon *Ddx3x* mRNA expression (*Figure 2C*). This indicates that *Ddx3y* sgRNAs are specific and effective.

We next assessed the requirement of *Ddx3y* for neurogenesis. In control E17.5 male brains (no sgRNA) GFP-positive cells were distributed fairly evenly across cortical bins (*Figure 2D, E*). Conversely, *Ddx3y* knockdown led to the accumulation of GFP+ cells in the VZ, with few GFP+ cells in the CP (*Figure 2D, E*). This defect could be due to impaired migration and/or altered production of neurons. Quantification of GFP+Sox2+ progenitors and GFP+Neurod2+ neurons showed that *Ddx3y* depletion led to significantly more progenitors and fewer neurons, as compared to control (*Figure 2F–H*). Importantly, these findings phenocopy acute *Ddx3x* knockdown by IUE in the embryonic brain (*Lennox et al., 2020*). This demonstrates that *Ddx3x* and *Ddx3y* have similar requirements for neurogenesis, suggesting that *Ddx3y* partially compensates for *Ddx3x* in cKO males. This finding may explain the divergent phenotypes of cKO male and female mice, as well as why some *DDX3X* mutations in human males are tolerated.

## *Ddx3x* cHet female and cKO male brains have more progenitors and fewer neurons

Because *DDX3X* syndrome females are heterozygous and males are hemizygous, we focused our analyses on cHet females and cKO males. Importantly, both genotypes have similar 30–40% reduction in *Ddx3x* mRNA relative to their sex-matched controls and normal brain size (*Figure 1*). Moreover, redundant functions of DDX3X and DDX3Y, along with transcriptional upregulation of *Ddx3y* in cKO males likely equalizes total DDX3 levels in cHet females and cKO males. This provides a rationale for evaluating both sexes to investigate requirements of *Ddx3x* in cortical development. For simplicity, going forward we collectively refer to the cHet females and cKO males as *Ddx3x* depleted.

Our data using both *Emx1*- and *Nex*-Cre drivers suggests that *Ddx3x* LoF impairs cortical development by specifically controlling progenitors. We thus quantified progenitors in *Emx1*-Cre, *Ddx3x* depleted brains at E13.5 and E14.5, stages at which both RGCs and IPs are abundant (*Figure 3*; *Figure 3—figure supplement 1*). Compared to E13.5 control mice, *Ddx3x* depleted brains showed similar numbers of mature IPs (Tbr2+Sox2-), although RGCs (Sox2+) trended higher (*Figure 3—figure supplement 1A–C*). However, by E14.5, the number of RGCs and mature IPs was significantly increased in *Ddx3x* depleted brains (*Figure 3A–C*). Moreover, there was a concomitant trend towards fewer Tbr2-Sox2- cells following *Ddx3x* depletion at both E13.5 and E14.5, suggesting potentially fewer neurons (*Figure 3—figure supplement 1D, E*). These alterations in cell composition did not significantly impact overall cortical thickness at E14.5, although there was a slight trend in reduced medial thickness (*Figure 3—figure supplement 1F*). These results are overall consistent with the lack of microcephaly in these mice (*Figure 1*).

We next assessed how these cell composition differences ultimately impact excitatory neuron number and laminar organization in P0 brains. The number of Tbr1 (Layer VI) and Ctip2 (Layer V) neurons was significantly reduced in *Ddx3x* depleted mice, and there was a trend towards reduced Lhx2 (Layer II/III) (*P*=0.078) (*Figure 3D–G*). However, laminar distribution was unaffected, suggesting that *Ddx3x* is largely dispensable for neuronal migration (*Figure 3—figure supplement 1G, H*). These data indicate that *Ddx3x* is essential for proper neuron number and suggests it is required across all stages of corticogenesis.

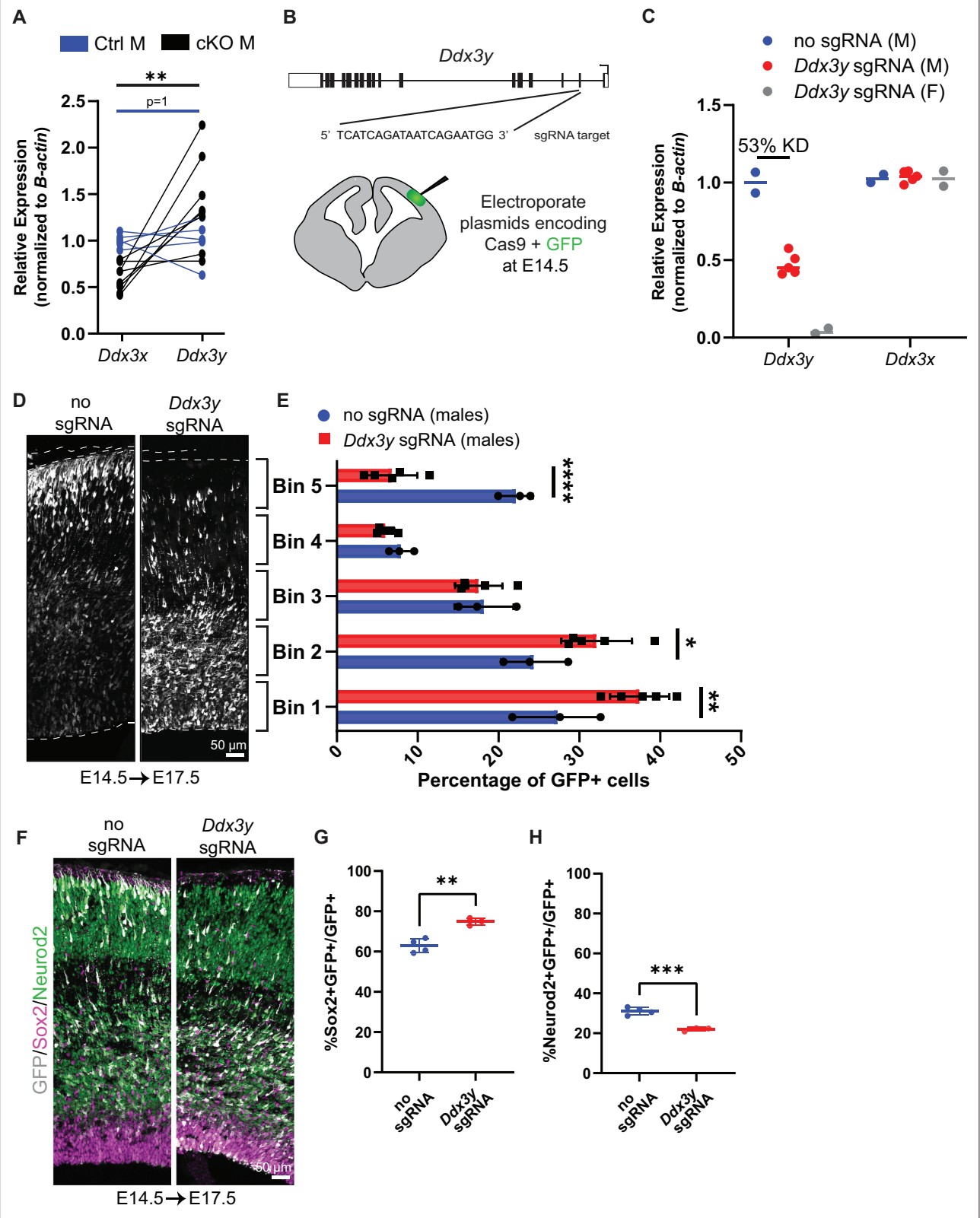

**Figure 2.** *Ddx3x* knockout is sexually dimorphic and *Ddx3y* phenocopies *Ddx3x* loss. (**A**) RT-qPCR quantification of *Ddx3x* and *Ddx3y* mRNA levels in FACS-isolated Tdtomato + cells from cKO male E11.5 cortices. n=5–8 embryos/condition. (**B**) Schematic of *Ddx3y* CRISPR sgRNA electroporation of E14.5 brain. (**C**) RT-qPCR quantification of *Ddx3y* and *Ddx3x* levels in GFP +FACS-isolated cells from E17.5 male and female mice electroporated with pCAG-GFP and either no sgRNA or *Ddx3y* sgRNA. n=2–5 embryos/condition. (**D**) Representative sections of E17.5 male brains electroporated at E14.5

*Figure 2 continued on next page*

*Figure 2 continued*

with pCAG-GFP and either no sgRNA or *Ddx3y* sgRNA and stained with anti-GFP (grey). Dotted lines, ventricular and pial surfaces; brackets delineate equivalently sized bins. (**E**) Quantification of distribution of GFP + cells. n=3–5 embryos/condition. (**F**) Same as (**D**), but sections were stained with anti-GFP (grey), Sox2 (magenta), and Neurod2 (green). (**G, H**) Quantification of GFP co-localization with Sox2 (**G**) or Neurod2 (**H**). n=3–5 embryos/condition. Scale bars, indicated. Error bars, S.D. *p<0.05, **p<0.01, ***p<0.001, ****p<0.0001. Student's paired, two-tailed t-test (**A**), Two-way ANOVA with Sidak's (**E**), Student's unpaired, two-tailed t-test (**G, H**).

## *Ddx3x* conditional heterozygous progenitors exhibit a longer cell cycle and undergo less neurogenic divisions

We next aimed to understand the cell biological mechanisms by which *Ddx3x* depletion impairs progenitor and neuron number. One possibility is that *Ddx3x* depleted progenitors re-enter the cell cycle rather than exiting and terminally differentiating. To investigate this, we quantified cell cycle exit in E14.5 *Ddx3x* depleted embryos, using a 24-hr pulse of the nucleotide analog, EdU, at E13.5. EdU+Ki67- cells have exited the cell cycle and terminally differentiated, whereas EdU+Ki67+ cells are still progressing through the cell cycle. At E14.5, there were significantly more EdU+Ki67+ cells in *Ddx3x* depleted brains relative to control (**Figure 4A–C**), indicating that *Ddx3x* is required for progenitor cell cycle exit.

In order to investigate whether *Ddx3x* depletion specifically impaired cell cycle exit of RGCs, IPs, or both, we quantified EdU+RGCs and IPs using Sox2 and Tbr2, respectively. We measured significantly more EdU+Sox2+ cells in *Ddx3x* depleted brains compared to control, indicating reduced cell cycle exit of RGCs (**Figure 4D**). We did not observe a change in mature IPs (**Figure 4—figure supplement 1A, B**). Moreover, there was a significant concomitant decrease in EdU+Sox2-Tbr2- cells (putative neurons) in *Ddx3x* depleted brains relative to control (**Figure 4E**), suggesting that *Ddx3x* depletion perturbs generation of neurons. Overall, these data demonstrate that *Ddx3x* depletion impairs cell cycle exit, predominantly in RGCs, resulting in generation of fewer excitatory neurons.

Based on these results, we posited that DDX3X controls neuron generation by impacting how progenitors divide. Indeed, DDX3X has a conserved role in regulating cell cycle in other contexts (**Chen et al., 2016a**; **Heerma van Voss et al., 2018**; **Kotov et al., 2016**; **Li et al., 2014**). To directly test whether *Ddx3x* depletion affects cortical progenitor cell cycle, we employed an established semi-cumulative labeling strategy (**Figure 4F**; **Quinn et al., 2007**). Briefly, at E14.5, EdU was injected intraperitoneally followed by a BrdU injection 1.5 hr later. At t=2 hr, embryonic brains were harvested and Ki67, EdU and BrdU were quantified. Strikingly, the *Ddx3x* depleted brains exhibited a significantly longer cell cycle (Tc) than control, nearly 1.5-fold longer (average 17 hr in ctrl vs 25 hr in cHet F/cKO M) (**Figure 4G and H**). However, *Ddx3x* depletion did not specifically impair S phase duration as Ts/Tc was not significantly different (**Figure 4I**). Further, there was no significant difference in the percentage of phospho-histone3 (PH3)+Sox2+ cells between controls and *Ddx3x* depleted brains (**Figure 4—figure supplement 1C, D**) indicating that G2/M length of RGCs is not specifically affected. Altogether, these data demonstrate that *Ddx3x* depletion prolongs overall cell cycle *in vivo*, without a clear bias towards any specific phase.

We next employed clonal live imaging to monitor both progenitor cell cycle duration and progeny generation (**Pilaz et al., 2016**). Primary cultures were produced from E14.5 cHet females, cKO males, and controls, and progenitors were live imaged (**Figure 5A**). After 24 hr, we fixed and stained cells to monitor direct progeny (**Figure 5A, B**). We did not observe any significant differences in mitosis duration (**Figure 4—figure supplement 1E**), which aligns with PH3 quantification at E14.5 (**Figure 4—figure supplement 1C, D**). Consistent with a prolonged cell cycle duration, *Ddx3x* depleted cells underwent significantly fewer re-divisions relative to controls (**Figure 5C–D**). These data are consistent with the longer cell cycle measured *in vivo*, demonstrating that progenitors *in vitro* reflect *in vivo* phenotypes.

Because *Ddx3x* depleted brains have a modest reduction in neurons, we also used live imaging to determine whether *Ddx3x* loss independently impairs the ability of progenitors to directly produce neurons. Following live imaging, cells were immunostained with Sox2, Tbr2, Tuj1 and Hoechst to discriminate between proliferative, asymmetric neurogenic, and symmetric neurogenic divisions (**Figure 5A, B**). *Ddx3x* depleted progenitors underwent significantly more proliferative (P,P) divisions and significantly fewer symmetric neurogenic (N,N) divisions relative to controls (**Figure 5E**). Asymmetric neurogenic (N,P) divisions were unchanged. Consistent with CC3 staining at E14.5, there were

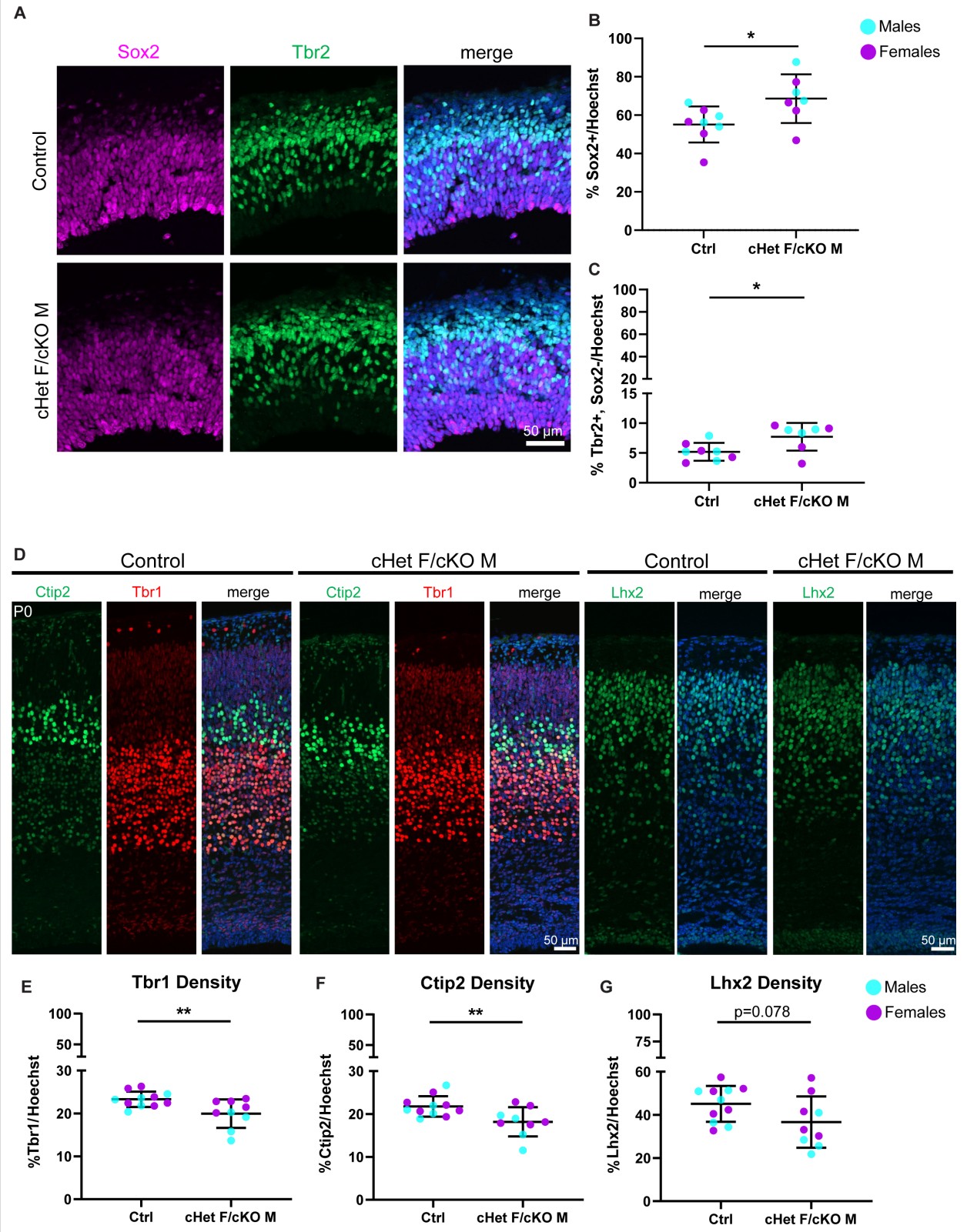

**Figure 3.** *Ddx3x* depletion leads to more RGCs and mature IPs, and fewer excitatory neurons across laminar layers. (**A**) Representative sections from E14.5 cortices stained with Sox2 (magenta) and Tbr2 (green) (control M and cKO M shown). (**B, C**) Quantification of density of Sox2+ (RGCs) (**B**) and Tbr2 +Sox2- (mature IPs) (**C**) cells relative to all cells (Hoechst) at E14.5. n=7–8 embryos/condition. (**D**) Representative sections stained with Ctip2 (green), Tbr1 (red), and Lhx2 (green) from P0 control and cHet F/cKO M cortices (control M and cKO M shown). (**E–G**) Quantification of laminar marker density

*Figure 3 continued on next page*

*Figure 3 continued*

for Tbr1 (**E**), Ctip2 (**F**), and Lhx2 (**G**) relative to all cells (Hoechst). n=8–10 embryos/condition. Scale bars, indicated. Error bars, S.D. *p<0.05, **p<0.01. Student's unpaired, two-tailed t-test (**B, C, E–G**).

The online version of this article includes the following figure supplement(s) for figure 3:

**Figure supplement 1.** *Ddx3x* depletion leads to more progenitors and less neurons at E13.5, but does not affect cortical thickness or laminar position of neurons.

virtually no apoptotic progeny in cHet females/cKO males and controls. These data demonstrate that, in addition to prolonged cell cycle, *Ddx3x*-deficient progenitors undergo more proliferative divisions and fewer symmetric neurogenic divisions.

Overall, these findings reveal that DDX3X controls neuron generation by acting in progenitors via two mechanisms. First, progenitor cell cycle duration is increased and progenitors are delayed in cell cycle exit, and second, progenitors that do exit the cell cycle tend to produce progenitors rather than neurons. This provides a mechanistic explanation for how *Ddx3x* controls cortical neuron generation.

## Ribosome profiling uncovers the translatome of E11.5 neural progenitors

We next sought to understand molecular mechanisms by which *Ddx3x* alters progenitor fate decisions and impairs neurogenesis. DDX3X is an RNA helicase with canonical requirements for translation initiation, particularly for mRNAs with structured 5′ UTRs (*Calviello et al., 2021*). Neural progenitors can be transcriptionally primed which is thought to promote generation of specific cell fates (*Hoye and Silver, 2021*; *Li et al., 2020*). However, a lack of genome-wide translational data in the developing cortex has limited our understanding of how translational control influences neurogenesis. To investigate translation at the earliest stages of cortical development and to identify which mRNAs require DDX3X for their translation, we performed ribosome profiling (Ribo-seq) (*Ingolia, 2016*) and RNA-seq using E11.5 microdissected cortices from cKO and control males and females (*Figure 6A*). To ensure maximal *Ddx3x* depletion and have the highest sensitivity for identifying DDX3X-dependent translation targets, we focused on cKO females and males. Further, we employed Ribo-seq at E11.5 to avoid confounds due to apoptosis in the cKO females. At this stage, the brain is also largely homogenous composed of mainly neural progenitors. We optimized Ribo-seq on E11.5 cortices (see Materials and methods) and performed extensive quality control to ensure that ribosome-protected fragments (RPFs) were the correct size (*Figure 6—figure supplement 1A, B*), mapped to the coding region as expected (*Figure 6—figure supplement 1C*), and were in the correct reading frame (*Figure 6—figure supplement 1D*; see Materials and methods).

We first examined translational regulation of wildtype progenitors. Overall, we observed a correlation between RNA and RPF abundance (Spearman $r$=0.9676) (*Figure 6—figure supplement 1E*). Using translation efficiency (TE, a metric reflecting translation per mRNA), taking into account both transcript levels (RNA-seq) and ribosome occupancy (Ribo-seq) (see Materials and methods), we assessed whether transcripts are translationally regulated in wildtype E11.5 progenitors. Towards this, we focused on TE of canonical mRNAs known to be enriched in RGCs, IPs, and deep (VI-V) and superficial layer (IV-II) neurons (*Di Bella et al., 2021*; *Telley et al., 2019*) (see *Supplementary file 3*) as compared to all other genes. We expected that RGC-enriched genes would be highly expressed and translated since RGCs are a predominant cell type at E11.5. IPs and deep layer neurons are produced beginning at E11.5 and E12.5, respectively. In contrast, superficial layer neurons are born later (E13.5-E17.5), and thus, we expected lower translation of these genes. To our surprise we found that at E11.5, RGC-, IP-, and deep layer neuron-enriched transcripts all had significantly higher TE than the average transcript expressed at this stage (other genes), consistent with translational upregulation (*Figure 6B*). Moreover, there was a notable divergence between deep (positive TE) and superficial (negative TE) layer neuron transcripts (*Figure 6B*), suggesting superficial layer neuron transcripts are translationally repressed. This suggests that in addition to transcriptional priming, early progenitors also use translational priming to direct cell fate. Moreover, these results indicate that observed differences in TE might reflect developmental stepwise translational repression.

Ribo-seq can also reveal the use of upstream open reading frames (uORFs), which frequently cause translational repression of the downstream, canonical ORF (*Johnstone et al., 2016*). We thus used

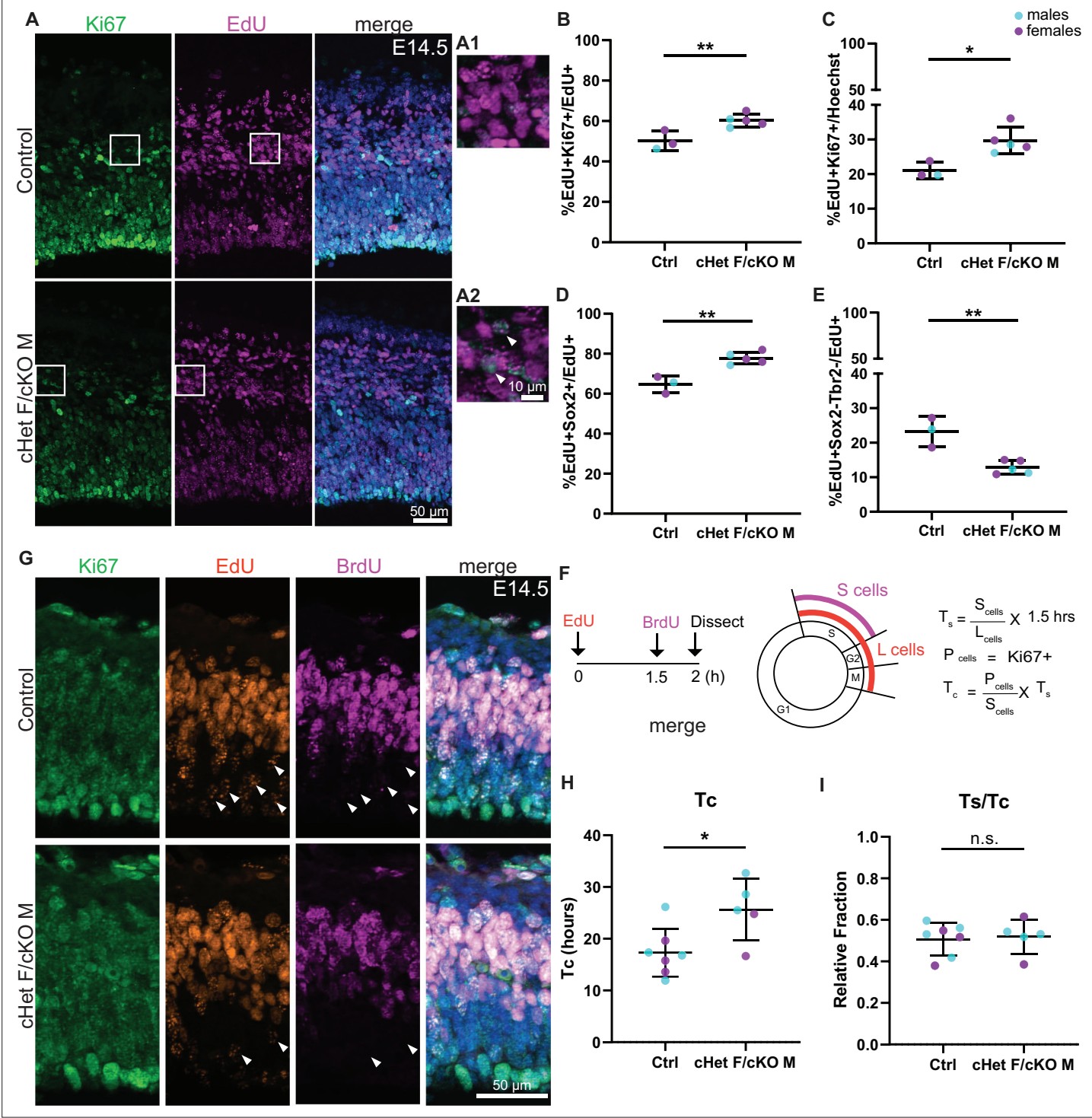

**Figure 4.** *Ddx3x* depletion impairs progenitor cell cycle exit and prolongs cell cycle duration. (**A**) Representative sections stained with Ki67 (green) and EdU (magenta) from E14.5 control and cHet F/cKO M mice (control F and cHet F shown) pulsed with EdU at E13.5, and higher magnification insets (A1, A2). (**B, C**) Quantification of Ki67 +EdU + relative to EdU + cells (**B**) and all cells (Hoechst, **C**). n=3–5 embryos/condition. (**D, E**) Quantification of EdU +Sox2+ (**D**) and EdU +Sox2-Tbr2- cells (**E**) relative to all EdU + cells. n=3–5 embryos/condition. (**F**) Schematic illustrating the semi-cumulative labeling paradigm and cell cycle formulas. *Figure 4F* has been adapted from Figure 4J from *Boyd et al., 2015*. (**G**) Representative medial sections of E14.5 control and cHet F/cKO M brains stained with Ki67 (green), EdU (red) and BrdU (magenta) and pulsed with EdU and BrdU (control M and cKO M shown). Arrows indicate EdU +BrdU cells (i.e: leaving cells). (**H**) Quantification of cell cycle duration (Tc) in control and cHet F/cKO M. n=5–7 embryos/

*Figure 4 continued on next page*

*Figure 4 continued*

condition. (**I**) Quantification of Ts/Tc in control and cHet F/cKO M. n=5–7 embryos/condition. Scale bars, indicated. Error bars, S.D. *p<0.05, **p<0.01. Student's unpaired, two-tailed t-test (**B–E, H, I**).

The online version of this article includes the following figure supplement(s) for figure 4:

**Figure supplement 1.** *Ddx3x* depletion prolongs cell cycle duration in RGCs and immature IPs but does not affect mitosis duration.

ORFquant to assess canonical ORFs and uORFs in cortical progenitors (*Calviello et al., 2020*). This revealed ~14,000 annotated ORFs and ~2500 uORFs, including a prominent uORF in the lissencephaly gene, *Pafah1b1* or *Lis1* (*Figure 6C*). We also identified ~1200 ORFs in non-coding RNAs, including a novel ORF in the *Rab26os* lncRNA (*Figure 6C*; *Figure 6—figure supplement 1F*, G). Thus, this rich dataset provides a valuable resource to interrogate the use of uORFs during cortical development and suggests an important mode of gene expression regulation in the developing cortex. Overall, these high-quality Ribo-seq data will enable the generation of novel hypotheses regarding translational control and cortical development.

## DDX3X-dependent translation targets are critical for neurogenesis

We then turned to our *Ddx3x* cKO data to discover DDX3X-dependent translation targets and identified 147 targets that had differential TE (p-adjusted <0.05) (*Figure 6D*). The low number of DDX3X

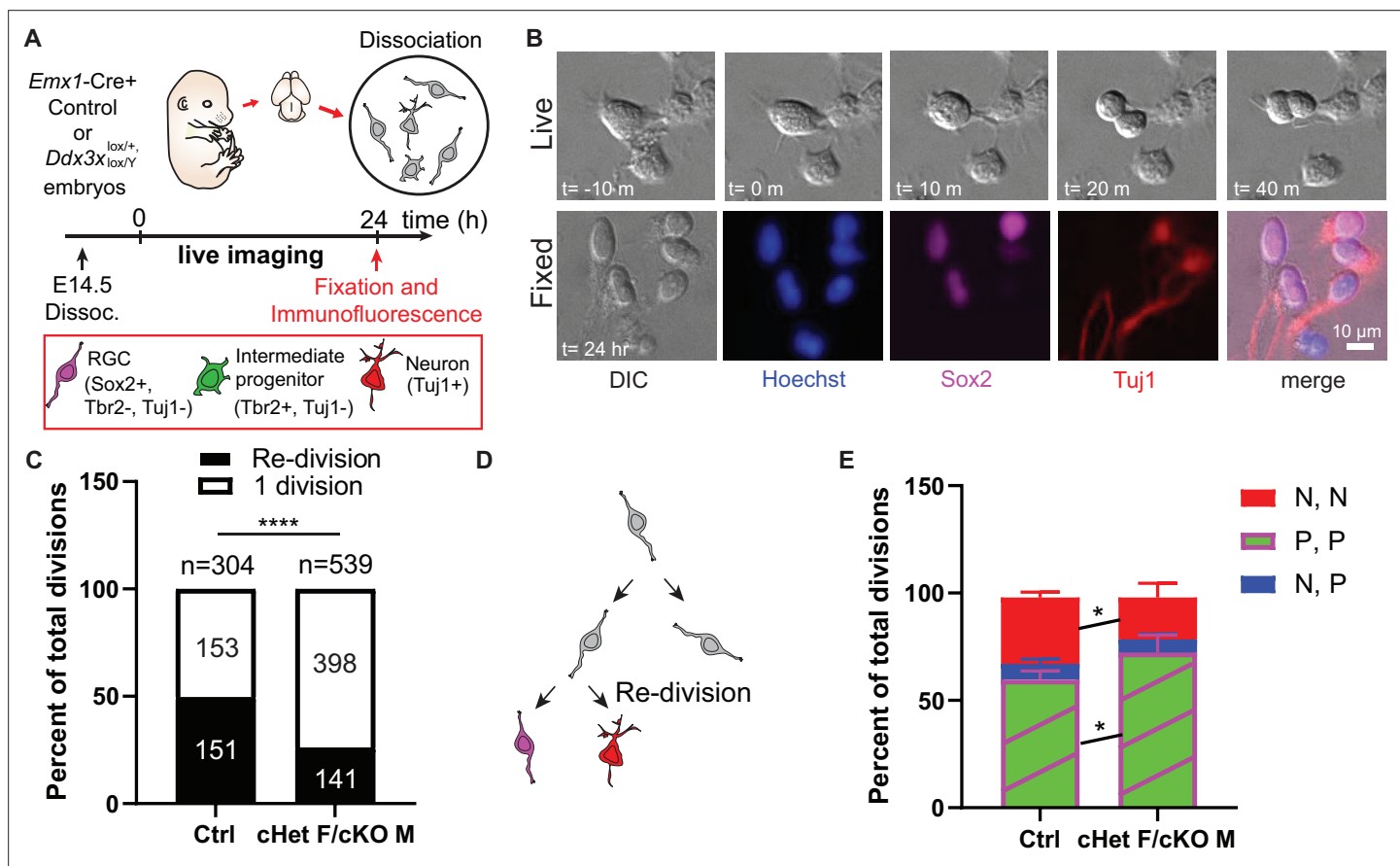

**Figure 5.** *Ddx3x*-depleted progenitors exhibit prolonged cell division and more proliferative divisions at the expense of neurogenic divisions. (**A**) Live imaging paradigm for monitoring cell fate. *Figure 5A* has been adapted from Figure 3A and E from *Pilaz et al., 2016*. (**B**) Live imaging DIC snapshots at indicated t=minutes or hours, and fixed images stained with indicated markers. (**C**) Quantification of re-divisions (black) and 1 division (white) in control and cHet F/cKO M. n=304 (control) and 539 (cHet F/cKO M) total cells. (**D**) Schematic illustrating an example of a re-division. (**E**) Quantification of cell fate for P,P divisions (2 Sox2+ RGCs, or 2 Tbr2+ IPs, or 1 Sox2+ RGC and 1 Tbr2+Tuj1- IP); P, N divisions (1 Tuj1+ neuron and either 1 Sox2+ RGC or 1 Tbr2+ IP); N, N divisions (2 Tuj1+ neurons). n=>70 cells/condition/trial with three trials. Scale bars, indicated. Error bars, S.D. *p<0.05, ****p<0.0001. Two-tailed Fisher's exact test (**C**), Two-way ANOVA with Sidak's correction (**E**).

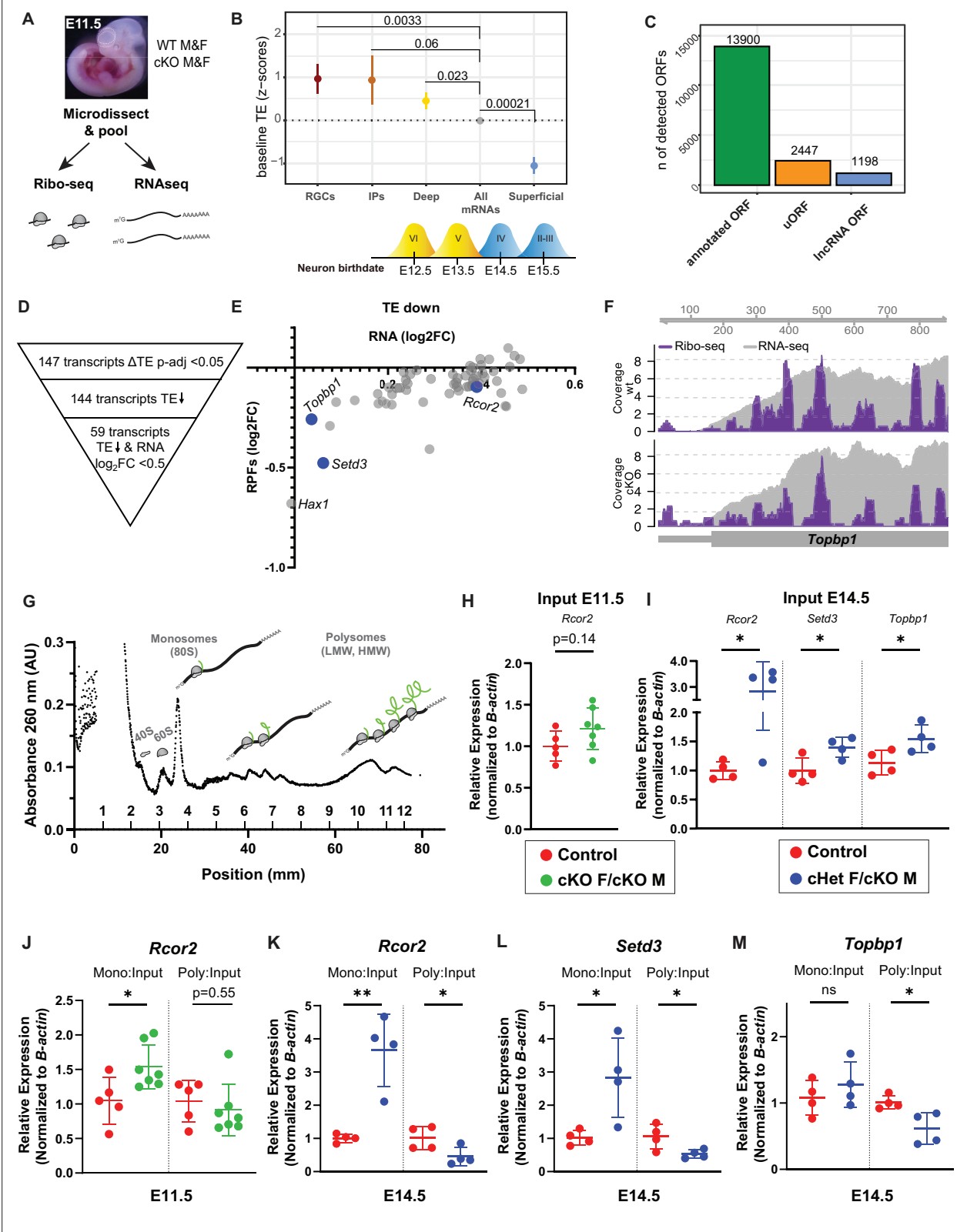

**Figure 6.** Ribosome Profiling in embryonic brains uncovers the E11.5 translatome including DDX3X-dependent translation targets. (**A**) Experimental paradigm for Ribo-seq and RNAseq of E11.5 cortices from control and cKO mice. n=3/sex/condition with four embryos pooled per n. (**B**) TE of transcripts enriched in RGCs, IPs, deep layer neurons (VI-V) and superficial layer neurons (IV-II) relative to all other mRNAs (TPM >10). Birthdates for laminar layers are indicated below. See ***Supplementary file 3*** for exact transcripts. (**C**) ORFquant analysis of wildtype Ribo-seq data showing

*Figure 6 continued on next page*

*Figure 6 continued*

identification of annotated ORFs and uORFs in protein-coding and non-coding isoforms. (**D**) Schematic illustrating how DDX3X-dependent targets were prioritized. (**E**) Scatter plot of RPFs log2FC versus RNA log2FC for 59 DDX3X-dependent targets with significantly lower TE. Putative Ribo-seq targets selected for validation are highlighted in blue. (**F**) IGV screenshots illustrating RNAseq reads (gray) and RPFs (Ribo-seq; purple) for *Topbp1* in cKO mice relative to control. (**G**) Representative trace from polysome fractionation of E14.5 cortical lysate. (**H–M**) RT-qPCR quantification of mRNA levels for Ribo-seq candidates in input samples at E11.5 (**H**) and at E14.5 (**I**), and monosome and polysome fractions at E11.5 (**J**) and E14.5 (**K–M**). n=5–7/condition (**H, J**) and 4/condition (**I, K–M**) with two embryos pooled per n. Error bars, S.D. *p<0.05, **p<0.01. Two-sided Wilcoxon test (**B**), Student's unpaired, two-tailed t-test (**H–M**).

The online version of this article includes the following source data and figure supplement(s) for figure 6:

**Figure supplement 1.** Quality Control Assessment of Ribosome Profiling in *Ddx3x* cKO mice.

**Figure supplement 1—source data 1.** Quality Control Assessment of Ribosome Profiling.

**Figure supplement 2.** Polysome fractionation and RNA immunoprecipitations showing DDX3X targets in the cortex.

**Figure supplement 2—source data 1.** RNA immunoprecipitations of targets by DDX3X.

translation targets is on-par with translational studies of DDX3X in immortalized cells and reinforces that DDX3X is not a general translation factor (*Calviello et al., 2021*). Virtually all targets (144 of 147) had a lower TE in cKO mice, consistent with DDX3X promoting translation. About half of these TE changes were driven by changes in the input RNA levels while the other half had significantly lower TE with little to no change in the corresponding input RNA. We thus focused on the 59 targets in which the input RNA was <0.5 log2FC, as these are most likely to be *bona fide* DDX3X translation targets (*Figure 6D, E*). *Rcor2*, *Setd3*, and *Topbp1*, were amongst those targets showing a general decrease in RPFs along the mRNA, adjusted p<0.05 (*Figure 6E, F*).

To orthogonally evaluate these DDX3X-dependent translational targets, we employed polysome fractionation using E11.5 cortices of cKO females and cKO males, as well as controls (*Figure 6G*). Therefore, we isolated RNA from the monosome and high molecular weight (HMW) polysome fractions and performed RT-qPCR of putative DDX3X translation targets. We observed that *Rcor2* and *Topbp1* mRNA were signficantly enriched in the monosome fraction of cKO females and males relative to controls, while the HMW polysome fraction was unaltered (*Figure 6J*, *Figure 6—figure supplement 2A*). Similar trends were seen with *Setd3* (*Figure 6—figure supplement 2B*). Of note, the input RNA levels of *Rcor2* trended to be elevated in cKO cortices as compared to controls (*Figure 6H*, *Figure 6—figure supplement 2A, B*), consistent with the Ribo-seq/RNA-seq results.

To further validate the translational targets and to assess targets that may reflect cell composition changes, we performed polysome fractionation at E14.5 again using cKO males and also cHet females (*Figure 6G*). We did not include E14.5 cKO females given the profound cell death evident at this timepoint (*Figure 1*). Both *Rcor2* and *Setd3* were significantly enriched in the monosome fraction and significantly depleted from the polysome fraction for cHet females/cKO males as compared to controls (*Figure 6K, L*), consistent with their decreased TE at E11.5. *Topbp1* was significantly depleted from the polysome fraction, but unchanged in the monosome fraction in cHet females/cKO males relative to controls (*Figure 6M*). Another putative target, *Hax1*, showed trends to shift from polysomes (*Figure 6—figure supplement 2C*). Interestingly, *Rcor2*, *Setd3,* and *Topbp1* were also significantly upregulated in the input RNA at E14.5 (*Figure 6I*) as compared to controls, even though their RNA levels were not significantly changed at E11.5. These shifts of DDX3X-dependent translation targets towards monosomes and concomitant depletion from polysomes were specific. For example, *β-actin* levels, were comparable between monosome fractions (*β-actin* CT average 25.06 vs 24.58) and polysome fractions (*β-actin* CT average 22.73 vs 22.72) of cHet females/cKO males and controls. Altogether the shift towards monosomes along with reduced TE reflects an essential requirement of DDX3X for translation of specific mRNAs in cortical progenitors *in vivo*.

We next asked whether DDX3X binds to these translational targets. Using RNA immunoprecipitations (RIP)(*Keene et al., 2006*), we transiently transfected N2A cells with GFP-hDDX3X and performed GFP pulldowns to isolate bound RNAs. RT-PCR for *Rcor2*, *Setd3*, and *Topbp1*, revealed specific binding to these transcripts by GFP-hDDX3X, but not in untransfected cells (*Figure 6—figure supplement 2D*).

We then queried whether any of these translation targets had known roles in neurogenesis. RCOR2 is critical for neural proliferation (*Monaghan et al., 2017*; *Wang et al., 2016*), and *Topbp1* deletion in

neural progenitors results in DNA damage and apoptosis (*Lee et al., 2012*). These requirements are notably similar to *Ddx3x* LoF, indicating that DDX3X-dependent translational targets are critical within neural progenitors for cortical development.

Another DDX3X-dependent target, *Setd3*, is expressed in the developing cortex (*Telley et al., 2019*), but its requirements for corticogenesis are unknown. SETD3 is a histidine methyltransferase, which may modify histone H3, as well as actin H73 (*Witecka et al., 2021*). Consistent with this, *Setd3*

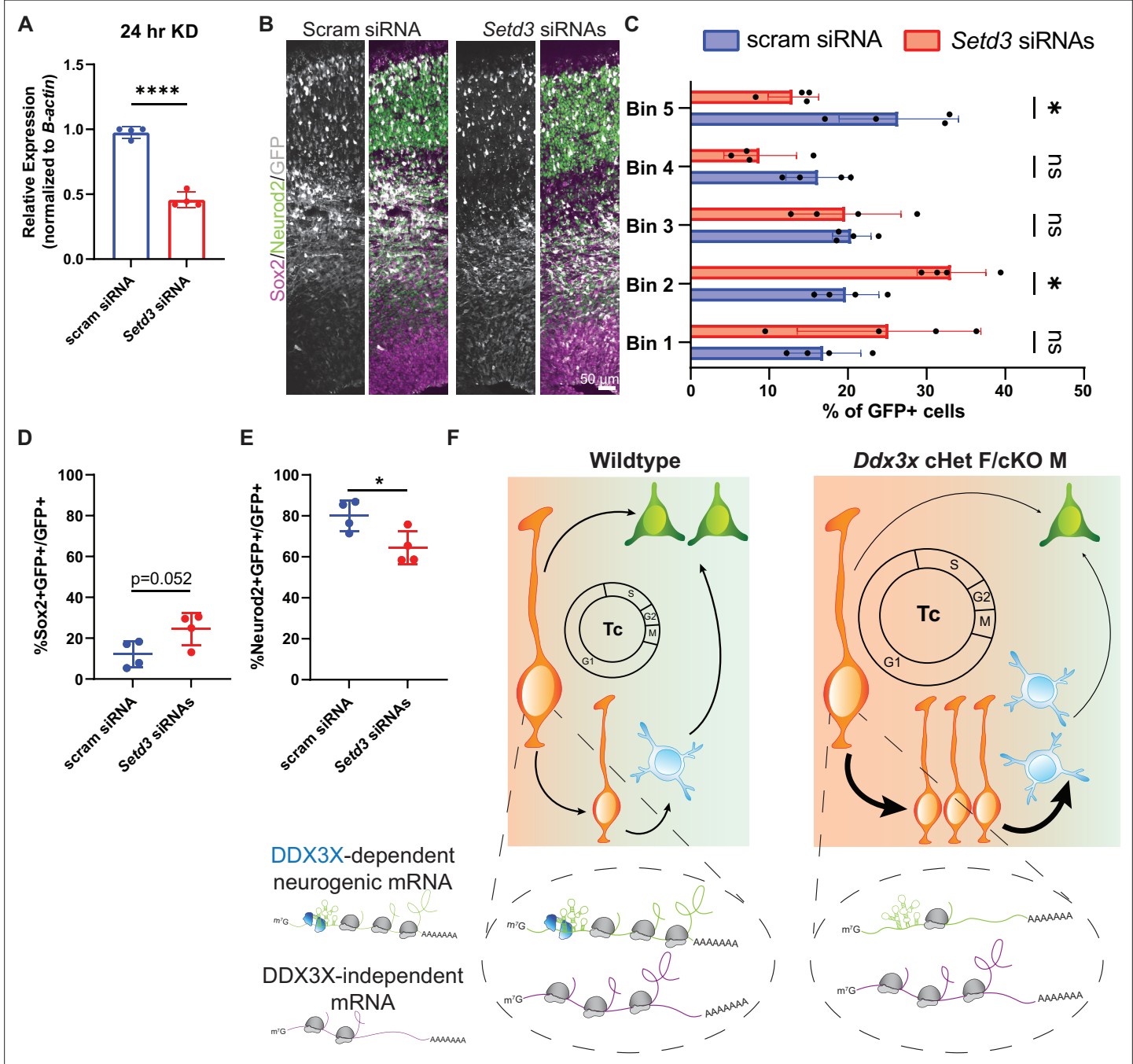

**Figure 7.** DDX3X-dependent translation target, *Setd3*, is required for neurogenesis. (**A**) RT-qPCR quantification of *Setd3* knockdown in N2A cells. n=4/ condition with two independent trials. (**B**) Representative sections of E17.5 brains from mice electroporated at E14.5 with pCAG-GFP and scrambled or *Setd3* siRNAs and immunostained with GFP (grey), Sox2 (magenta), Neurod2 (green). (**C**) Quantification of distribution of GFP-positive cells in 5 even bins of cortex. n=4 embryos/condition. (**D, E**) Quantification of GFP co-localization with Sox2+ (**D**) and Neurod2+ (**E**) cells. n=4 embryos/condition. (**F**) Schematic model summarizing how loss of DDX3X-dependent translation impairs neurogenesis. Scale bars, indicated. Error bars, S.D. *p<0.05, ****p<0.0001. Student's unpaired, two-tailed t-test (**A, D, E**), Two-way ANOVA with Sidak's correction (**C**).

is implicated in cancer progression and cell cycle control, as well as cytoskeletal integrity (*Witecka et al., 2021*). To test the requirements of *Setd3* for cortical development, we validated siRNAs by knockdown *Setd3* in N2A cells and observed a ~50% reduction after 24 hours (*Figure 7A*). We then performed IUEs to deliver either scrambled or *Setd3* siRNAs, along with pCAG-GFP, at E14.5 and collected embryos at E17.5. *Setd3* knockdown significantly altered the distribution of GFP + cells, with significantly more cells in the VZ and fewer cells in the CP (*Figure 7B, C*). Moreover, *Setd3* knockdown resulted in more Sox2+ RGCs and fewer Neurod2+ neurons (*Figure 7D, E*), phenocopying *Ddx3x* LoF. This suggests that decreased TE of *Setd3,* as well as other DDX3X-dependent targets, may synergistically contribute to *Ddx3x* LoF neurogenesis phenotypes (*Figure 7F*). Taken together these data show that DDX3X controls translation of a repertoire of genes which are collectively critical in neural progenitors for neurogenesis.

## Discussion

Mutations in *DDX3X* are a leading cause of ID, as well as other developmental phenotypes, which are classified as *DDX3X* syndrome (*Beal et al., 2019*; *Johnson-Kerner et al., 1993*; *Lennox et al., 2020*; *Snijders Blok et al., 2015*; *Wang et al., 2018*). Half of the over 100 known *DDX3X* mutations are nonsense, predicted to act in a LoF manner. Yet, the molecular and cellular mechanisms by which *Ddx3x* LoF perturbs cortical development have been largely unknown. Here, we leveraged a new genetic model of *Ddx3x* LoF to discover that DDX3X promotes neuronal generation by regulating the length and fate of progenitor divisions, namely RGCs and IPs. We further use *in vivo* Ribo-seq to define the repertoire of transcripts whose translation depends upon DDX3X, many of which are critical for neurogenesis. Our results provide invaluable new insights into the underpinnings of *DDX3X* syndrome and biology, and highlight candidate targets and possible pathways for therapeutic intervention.

### *Ddx3x* dosage underlies cell fate and sexually dimorphic phenotypes

While the vast majority of *DDX3X* syndrome individuals are female, an increasing number of male individuals have been identified (*Kellaris et al., 2018*; *Nicola et al., 2019*), comprising ~5% of all cases (ddx3x.org). Using our conditional mouse model, we interrogated phenotypic overlap in males and females, as well as the role of the paralog *Ddx3y*. Complete loss of *Ddx3x* led to microcephaly in females, but not in hemizygous males, suggesting that brain development is sexually dimorphic. We posit that *Ddx3y* expression explains why cKO male mice are phenotypically milder than cKO females; we found that *Ddx3y* loss phenocopies *Ddx3x* neurogenesis defects. Likewise, in both the hindbrain and hematopoietic system, *Ddx3y* can also compensate for loss of *Ddx3x in vivo* (*Patmore et al., 2020*; *Szappanos et al., 2018*). The ability of *Ddx3y* to compensate for *Ddx3x* loss may be due in part to redundant translational regulation (*Venkataramanan et al., 2021*), as well as transcriptional adaptation of *Ddx3y* in response to *Ddx3x* loss (*Figure 2*). Taken together these data provide a possible explanation for why males carrying *DDX3X* mutations are viable (*Kellaris et al., 2018*; *Lennox et al., 2020*; *Nicola et al., 2019*). Because *Ddx3x* partially escapes XCI in females (*Figure 1*), we hypothesize that DDX3Y normalizes total DDX3 levels between males and females; this may explain why cHet females and cKO males have comparable phenotypes.

In addition to being sexually dimorphic, we discover that neural cells of the developing cortex are sensitive to *Ddx3x* knockout, as loss of one versus two copies causes vastly different corticogenesis phenotypes. In the case of cKO females, there is profound apoptosis in progenitors and neurons, whereas cHet females and cKO males have impaired neurogenesis without cell death. This sensitivity of the developing brain to *Ddx3x* is highly relevant for interpreting the spectrum of mutations annotated as nonsense. For example, there may be nonsense mutations that act as a pure LoF (i.e.: an N-terminal mutation that undergoes NMD) as well as mutations that are hypomorphic (i.e.: a C-terminal mutation that escapes NMD and might have some functional protein activity).

### *Ddx3x* is required for neural progenitor cell division which underlies cortical abnormalities

We show that DDX3X controls the function of neural progenitors and generation of excitatory neurons throughout cortical development through two key mechanisms. First, *Ddx3x* depletion increases progenitor cell cycle duration by 1.4-fold. It does so without altering any specific phase, consistent

with a study of DDX3X in cancer cells using FUCCI (*Heerma van Voss et al., 2018*). Additionally, *Ddx3x* depleted progenitors *in vitro* underwent fewer re-divisions in a 24-hr period; if this increase in cell cycle duration were sustained over the course of neurogenesis, we predict it would result in overall fewer progenitor divisions, and ultimately fewer neurons (*Takahashi et al., 1995*). Consistent with this, neurons were modestly reduced across laminar layers at P0 in *Ddx3x* mutants. Second, *Ddx3x* controls the balance of proliferative versus neurogenic divisions, resulting in reduced generation of neurons. This is particularly fascinating as it demonstrates that DDX3X functions in progenitors to direct neural fates of daughter progeny. Our Ribo-seq suggests that these cellular mechanisms might be influenced by translational control in progenitors.

Going forward, it will be important to further dissect how DDX3X dosage influences neural progenitor cell cycle and cell fate. *Ddx3x* depletion had a particularly potent impact on RGCs, relative to IPs, suggesting that RGCs may be especially sensitive to loss of DDX3X, which is consistent with higher *Ddx3x* expression in RGCs relative to IPs (*Telley et al., 2019*). Moreover, this could reflect differences in cell cycle duration of these progenitors (*Arai et al., 2011*).

Using a conditional mouse model provided distinct advantages to propel our understanding of *Ddx3x* requirements during cortical development. New insights into *Ddx3x* dosage requirements and redundant roles of DDX3X and DDX3Y would not have been gained with germline LoF models, due to male lethality (*Chen et al., 2016a*). A recent report assessed behavioral phenotypes of a germline mouse model of *Ddx3x* haploinsufficiency in females (*Boitnott et al., 2021*). Consistent with our cHet phenotypes, these females were not grossly microcephalic, but did have reduced Ctip2+ (layer V) neurons in the somatosensory cortex at P3, which persisted into adulthood. These germline mice also have overall gross delay, defects in motor coordination, and sensory behavioral abnormalities. Our study thus provides valuable insights to explain the embryonic underpinnings of these behavioral deficits. Finally, while our analyses cannot rule out subtle brain size defects in cHet females, overall our findings are consistent with the human syndrome where the microcephaly is mild or absent in individuals with nonsense mutations (*Lennox et al., 2020*).

We also gained new insights into cell-autonomous *Ddx3x* requirements in brain development. Using a *Nex*-Cre cKO mouse model, we found that *Ddx3x* is largely dispensable in newborn neurons for their migration and survival, indicating that observed neuronal loss largely arises from impairments in neural progenitors. However, *Ddx3x* does control neurite outgrowth (*Chen et al., 2016b*) and is a component of neuronal transport granules (*Elvira et al., 2006*; *Kanai et al., 2004*), suggesting it could function in mature neurons. DDX3X likely functions in other CNS cell types, as well, as 50% of *DDX3X* syndrome individuals present with white matter loss (*Lennox et al., 2020*), which could arise from impaired gliogenesis. Indeed, Perturb-seq of 30 ASD-risk genes found that *Ddx3x* depletion alters oligodendrocyte gene expression (*Jin et al., 2020*). Moreover, *DDX3X* also regulates innate immunity (*Szappanos et al., 2018*), raising the intriguing idea that *Ddx3x* could function in microglia to influence cortical development. Thus, roles for *Ddx3x* in glial cells and mature neurons should be explored further.

## Translation during cortical development and DDX3X-dependent translational control of progenitors

In the neurodevelopment field there is a wealth of transcriptomic data, but far less translation/proteomic data for embryonic mouse cortex. Using Ribo-seq, we have generated a valuable new resource for understanding translation at the onset of neurogenesis. To our knowledge, this is one of the first reports to assess wildtype translation using Ribo-seq at the onset of neurogenesis, with exception of a recent pre-print which examined later stages of development (*Harnett et al., 2021*). scRNAseq studies have revealed that transcripts important for deep and superficial layer neuronal fates are expressed in RGCs *Telley et al., 2019*; however, it's generally thought that these are translationally repressed until neurons are born. Intriguingly, our data show that deep layer transcripts have a higher TE than superficial layer transcripts, approaching that of RGCs and IPs. This suggests that there could be translational priming of progenitors, in addition to transcriptional priming. Given the instrumental role that translational factors play in cortical development (*Hoye and Silver, 2021*; *Kraushar et al., 2014*; *Yang et al., 2014*; *Zahr et al., 2018*), our data thus provides valuable targets which can be functionally tested at neuroepithelial stages and beyond.

Although DDX3X is a known translation regulator (*Calviello et al., 2021*; *Oh et al., 2016*), a key gap has been the identification of its targets in the developing brain. Using Ribo-seq, we discovered DDX3X-dependent targets in neural progenitors *in vivo*. Both *Rcor2* and *Topbp1* have known roles in cortical neurogenesis and we further established new requirements for *Setd3*. Both *Rcor2* and *Setd3* are epigenetic regulators (*Wang et al., 2016*), suggesting their decreased TE may alter gene expression and cell fate. Notably, epigenetic regulators can influence gene expression by complexing with transcription factors. Indeed, RCOR2 binds to Insulinoma-associated 1 (*Monaghan et al., 2017*), a transcription factor required in IPs for neurogenesis (*Farkas et al., 2008*). DDX3X-mediated epigenetic regulation will be fascinating to investigate in future studies, especially considering that many epigenetic regulators and chromatin-modifying enzymes are implicated in ID and ASD (*Grafodatskaya et al., 2010*). In sum, our translational analysis highlights new downstream pathways relevant for DDX3X function and disease pathology.

We speculate that reduced TE of DDX3X-dependent transcripts collectively contributes to phenotypes in mouse models and perhaps in human *DDX3X* syndrome. Given that *Ddx3x* LoF modestly affects TE of many transcripts in neural progenitors, we do not anticipate that increasing TE of any one transcript would alleviate *Ddx3x* LoF phenotypes. However, understanding how DDX3X is recruited to these transcripts to specifically promote translation initiation might uncover valuable mechanisms by which transcripts with low TE could be boosted translationally for therapeutic intervention. Likewise, elucidating RNA structures of these targets might lead to development of therapeutic small molecules which can overcome DDX3X-dependent translation.

Interestingly, DDX3X-dependent translation targets were shifted from polysomes to monosomes, consistent with reduced TE and a role for DDX3X in translational initiation. However, the basal RNA expression of DDX3X targets was also higher at E14.5 compared to E11.5. This could be explained by developmental differences in DD3X levels. Alternatively, it could suggest that reduced TE of DDX3X-dependent targets leads to a feedback mechanism whereby these mRNAs are transcriptionally upregulated over time. This potential feedback loop is relevant for those pursuing Ribo-seq experiments to consider, as putative targets are typically defined by RPF density changes but not the input RNA (*Ingolia, 2016*).

### The landscape of DDX3X mutations with divergent phenotypes

Our work further substantiates the importance of post-transcriptional RNA regulation in cortical development and disease. There is a strong association between mutations in translation regulators and neurodevelopmental disorders (*Chen et al., 2019*; *Hoye and Silver, 2021*). As we have shown, investigation of DDX3X and translational regulators can give valuable insights into the etiology of these disorders by discovering networks of neurogenesis regulation including crucial factors such as SETD3.

Our work also provides a mechanistic understanding of how *Ddx3x* LoF impairs embryonic cortical development; this may model human *DDX3X* nonsense mutations that result in haploinsufficiency. The extent to which these mechanisms extend to all mutations is unknown. Indeed, it is imperative to define how the full spectrum of *DDX3X* mutations impact its expression and ultimately, influence corticogenesis. Half of *DDX3X* mutations are missense, with a subset showing more clinically severe outcomes relative to those carrying nonsense mutations (*Lennox et al., 2020*). Interestingly, *DDX3X* missense mutations might differentially impair translation of DDX3X targets (*Calviello et al., 2021*; *Lennox et al., 2020*). Thus, diverse *DDX3X* mutations may result in divergent molecular and cellular pathologies and our study provides an important foundation upon which future investigations of *DDX3X* mutations may be compared.

## Materials and methods

**Key resources table**

| Reagent type (species) or resource | Designation | Source or reference | Identifiers | Additional information |
| --- | --- | --- | --- | --- |
| Gene (*Mus musculus*) | *Ddx3x* | NA | MGI:103064; NCBI Gene: 13,205 | |
| Gene (*M. musculus*) | *Ddx3y* | NA | MGI:1349406; NCBI Gene: 26,900 | |

*Continued on next page*

*Continued*

| Reagent type (species) or resource | Designation | Source or reference | Identifiers | Additional information |
|---|---|---|---|---|
| Gene (*M. musculus*) | *Rcor2* | NA | MGI:1859854; NCBI Gene: 104,383 | |
| Gene (*M. musculus*) | *Setd3* | NA | MGI:1289184; NCBI Gene: 52,690 | |
| Gene (*M. musculus*) | *Topbp1* | NA | MGI:1920018; NCBI Gene: 235,559 | |
| Genetic reagent (*M. musculus*) | C57BL/6 J | Jackson Laboratory | JAX #000664; RRID:IMSR_JAX:000664 | |
| Genetic reagent (*M. musculus*) | *Emx1*-Cre | Jackson Laboratory | JAX #005628; RRID:IMSR_JAX:005628; MGI:2684610 | MGI symbol: Emx1 tm1(cre)Krj |
| Genetic reagent (*M. musculus*) | *Rosa26*<sup>Ai14</sup> | Jackson Laboratory | JAX #007914; RRID:IMSR_JAX:007914 | MGI symbol: Gt(ROSA)26Sor tm14(CAG-tdTomato)Hze |
| Genetic reagent (*M. musculus*) | *Neurod6-Cre (NEX*-Cre) | PMID:17146780 | MGI:2668659 | MGI symbol: Neurod6 tm1(cre)Kan |
| Genetic reagent (*M. musculus*) | *Ddx3x*<sup>lox/lox</sup> | PMID:27179789 | MGI:5774968 | MGI symbol: Ddx3x tm1.1Lyou |
| Genetic reagent (*M. musculus*) | *Ddx3y* sgRNAs | this paper | NCBI gene: 26,900 | generated with Benchling for depleting *Ddx3y*; see *Figure 2* |
| Genetic reagent (*M. musculus*) | smFISH probes for *Ddx3x* | this paper | NCBI gene: 13,205 | generated with script from *Tsanov et al., 2016* for monitoring *Ddx3x* RNA; see *Figure 1* |
| Genetic reagent (*M. musculus*) | Scrambled siRNAs | Qiagen | Qiagen:1022076 | |
| Genetic reagent (*M. musculus*) | *Setd3* siRNAS | Qiagen | Qiagen:1027416 | Gene ID: 52,690 |
| Cell line (*M. musculus*, male) | Neuro-2a | ATCC | ATCC:CCL-131; RRID:CVCL_0470 | |
| Strain, strain background | NEB 5-alpha Competent *E. coli* (High Efficiency) | New England Biolabs | NEB:C2987H | |
| Sequence-based reagent (*M. musculus*) | *Rcor2* qPCR primers | Harvard PrimerBank; PMID:22086960 | | Harvard PrimerBank ID: 154147710 c2; Forward 5'-TGCTTCTGTGGCATAAACACG-3'; Reverse 5'-GGCTGGGAATCACCTTGTCAG-3' |
| Sequence-based reagent (*M. musculus*) | *Setd3* qPCR primers | Harvard PrimerBank; PMID:22086960 | | Harvard PrimerBank ID: 21312266a1; Forward 5'-AAATCAGGTACTGGGGCTACA-3'; Reverse 5'-GGCCCATTTCATTAGATCAGGGA-3' |
| Sequence-based reagent (*M. musculus*) | *Topbp1* qPCR primers | Harvard PrimerBank; PMID:22086960 | | Harvard PrimerBank ID: 118130322 c1; Forward 5'-CAGGATTGTTGGTCCTCAAGTG-3'; Reverse 5'-ACAGGATACAGTTACGTCAGACA-3' |
| Antibody | anti-SOX2 (rat monoclonal) | ThermoFisher | ThermoFisher:14-9811-82; RRID:AB_11219471 | (1:1000) |
| Antibody | anti-BrdU (rat monoclonal) | Abcam | Abcam:ab6326 | (1:200) |
| Antibody | anti-TUJ1 (mouse monoclonal) | Biolegend | Biolegend:801202; RRID:AB_10063408 | (1:2000) |
| Antibody | anti-CTIP2 (rat monoclonal) | Abcam | Abcam:AB18465 | (1:500) |
| Antibody | anti-TBR2 (rabbit polyclonal) | Abcam | Abcam:AB23345; RRID:AB_778267 | (1:1000) |
| Antibody | anti-CC3 (rabbit polyclonal) | Cell Signaling | Cell Signaling:9661; RRID:AB_2341188 | (1:250) |
| Antibody | anti-NEUROD2 (rabbit polyclonal) | Abcam | Abcam:AB104430; RRID:AB_10975628 | (1:500) |
| Antibody | anti-Ki67 (rabbit monoclonal) | Cell Signaling Technology | Cell Signaling:12,202 | (1:1000) |
| Antibody | anti-PH3 (rabbit polyclonal) | Millipore | Millipore:06–570 | (1:500) |

*Continued on next page*

*Continued*

| Reagent type (species) or resource | Designation | Source or reference | Identifiers | Additional information |
|---|---|---|---|---|
| Antibody | anti-TBR1 (rabbit monoclonal) | Cell Signaling Technology | Cell Signaling Technology:49,661 S | (1:1000) |
| Antibody | anti-DDX3X (rabbit polyclonal) | Sigma Aldrich | Sigma Aldrich:HPA001648; RRID:AB_1078635 | (IF, 1:500; western, 1:1000) |
| Antibody | anti-GFP (chicken polyclonal) | Abcam | Abcam:Ab13970; RRID:AB_300798 | (1:1000) |
| Antibody | anti-B-actin (mouse monoclonal) | Santa Cruz | Santa Cruz:sc-47778 | (1:500) |
| Antibody | anti-mouse HRP (goat polyclonal) | ThermoFisher | ThermoFisher:32430; RRID:AB_1185566 | (1:2000) |
| Antibody | anti-rabbit HRP (goat polyclonal) | ThermoFisher | ThermoFisher:A16110; RRID:AB_2534782 | (1:2000) |
| Antibody | anti-GFP (mouse monoclonal) | Santa Cruz | Santa Cruz:sc9996 | |
| Antibody | AlexaFluor-conjugated secondary antibodies (488, 555, 568, 594, 647) | ThermoFisher | | (1:500) |
| Recombinant DNA reagent | pX330-U6-Chimeric_BB-CBh-hSpCas9 | Addgene | AddGene:42230; RRID:Addgene_42230 | |
| Recombinant DNA reagent | pX330-U6-Chimeric_BB-CBh-hSpCas9+*Ddx3y* guides | this paper | | Cloned for depletion of *Ddx3y*; see **Figure 2** |
| Recombinant DNA reagent | pCAG-GFP | PMID:32135084 | | |
| Recombinant DNA reagent | pCAG-GFP-human DDX3X | PMID:32135084 | | |
| Commercial assay or kit | Click-it EdU AlexaFluor 594 imaging kit | Life Technologies | Life Technologies:c10339 | |
| Commercial assay or kit | Qiagen RNAeasy kit | Qiagen | Qiagen:74,034 | |
| Commercial assay or kit | Qiagen miRNA library prep kit | Qiagen | Qiagen:331,502 | |
| Commercial assay or kit | RiboMinus Eukaryote kit v2 | ThermoFisher | ThermoFisher:A15020 | |
| Commercial assay or kit | RNA Analysis Kit (15 nt) | Agilent | Agilent:DNF-471 | |
| Commercial assay or kit | Kapa mRNA HyperPrep kit with mRNA capture | KapaBiosystems | KapaBiosystems:KR1352 | |
| Commercial assay or kit | iScript cDNA synthesis kit | BioRad | BioRad:1708891 | |
| Commercial assay or kit | iTaq Universal Sybr Green Supermix | BioRad | BioRad:1725121 | |
| Commercial assay or kit | BCA protein quantification | ThermoFisher | ThermoFisher:23,227 | |
| Commercial assay or kit | ECL | ThermoFisher | ThermoFisher:32,106 | |
| Chemical compound, drug | EdU | ThermoFisher | ThermoFisher:A10044 | |
| Chemical compound, drug | BrdU | Sigma Aldrich | Sigma:B5002 | |
| Chemical compound, drug | Cycloheximide | Calbiochem | Sigma:239,764 | |

*Continued on next page*

*Continued*

| Reagent type (species) or resource | Designation | Source or reference | Identifiers | Additional information |
|---|---|---|---|---|
| Software, algorithm | Fiji/ImageJ | PMID:22743772 | | v1.52i |
| Software, algorithm | QuPATH | PMID:29203879 | | v0.3.2 |
| Software, algorithm | bowtie2 | PMID:22388286 | | v2.4.4 |
| Software, algorithm | STAR | PMID:23104886 | | v2.7.9a |
| Software, algorithm | RibosomeProfilingQC | DOI:10.18129/B9.bioc. ribosomeProfilingQC | | v1.8.0 |
| Software, algorithm | Ribo-seQC | DOI:10.1101/601468 | | v0.99 |
| Software, algorithm | DESeq2 | PMID:25516281 | | v1.34.0 |
| Software, algorithm | ORFquant | PMID:33765284 | | v1.02 |
| Software, algorithm | R | R Foundation for Statistical Computing | | v4.1.0 |
| Other | Vectashield | Vector Labs | Vector Labs:H-1000–10 | See immunoflourescence section in methods |
| Other | DAPI stain | ThermoFisher | ThermoFisher:D1306 | See FACS section in methods |
| Other | Propidium iodide stain | ThermoFisher | ThermoFisher:P3566 | See FACS section in methods |
| Other | Hoechst stain | ThermoFisher | ThermoFisher:H3570 | See immunoflourescence section in methods |
| Other | Turbo DNase I | Invitrogen | Invitrogen:AM2238 | See RNAseq and ribosome footprinting section in methods |
| Other | Superase In | Invitrogen | Invitrogen:AM2694 | See RNAseq and ribosome footprinting section in methods |
| Other | TRIzol | Invitrogen | Invitrogen:15596026 | See RNAseq and ribosome footprinting section in methods |
| Other | GlycoBlue | Invitrogen | Invitrogen: AM9515 | See RNAseq and ribosome footprinting section in methods |
| Other | PNK enzyme | New England Biolabs | NEB:M0247S | See RNAseq and ribosome footprinting section in methods |
| Other | 15% acrylamide denaturing urea-gel | BioRad | BioRad:4566053 | See RNAseq and ribosome footprinting section in methods |
| Other | 2 X sample dye | Novex | Novex:LC6876 | See RNAseq and ribosome footprinting section in methods |
| Other | dsDNA ladder | ThermoFisher | ThermoFisher:10488023 | See RNAseq and ribosome footprinting section in methods |
| Other | miRNA ladders | New England Biolabs | NEB:N2102S | See RNAseq and ribosome footprinting section in methods |
| Other | SYBR gold | Invitrogen | Invitrogen:S11494 | See RNAseq and ribosome footprinting section in methods |
| Other | SpinX column | Corning | Corning:CLS8162 | See RNAseq and ribosome footprinting section in methods |
| Other | TRIzol LS reagent | ThermoFisher | ThermoFisher:10296010 | See RNA Immunoprecipitation section in methods |
| Other | Lipofectamine 2000 | ThermoFisher | ThermoFisher:11668019 | See RNA Immunoprecipitation section in methods |
| Other | Protein G-coated Dynabeads | ThermoFisher | ThermoFisher:0003D | See RNA Immunoprecipitation section in methods |
| Other | GoTaq Green Master Mix | Promega | Promega:M712 | See RNA Immunoprecipitation section in methods |
| Other | 1 X RIPA buffer | Pierce | Pierce:89,900 | See SDS-PAGE and western blot analysis section in methods |
| Other | 2 X sample buffer | BioRad | BioRad:1610737 | See SDS-PAGE and western blot analysis section in methods |

*Continued*

| Reagent type (species) or resource | Designation | Source or reference | Identifiers | Additional information |
|---|---|---|---|---|
| Other | protease inhibitors | Sigma Aldrich | Sigma:78,429 | See SDS-PAGE and western blot anlaysis section in methods |
| Other | 12% polyacrylamide gel | BioRad | BioRad:4568046 | See SDS-PAGE and western blot anlaysis section in methods |
| Other | PVDF membrane | BioRad | Biorad:1704157 | See SDS-PAGE and western blot anlaysis section in methods |

## Mouse husbandry

All animal use was approved by the Duke Division of Laboratory Animal Resources and the Institutional Animal Care and Use Committee. The following lines were used and genotyped as described, all on C57BL/6 J background: *Emx1*-Cre (005628) (*Gorski et al., 2002*) and *Rosa26^{Ai14}* (007914) (*Madisen et al., 2010*) (Jackson Laboratory); *Neurod6*-Cre (*Nex*-Cre) (*Goebbels et al., 2006*) (gift, Klaus-Nave); *Ddx3x^{lox/lox}* (*Chen et al., 2016a*) (gift, Li-Ru You). Plug dates were defined as E0.5 on the morning the plug was identified.

## Statistical methods and rigor

Exact statistical tests, p-values, and n for each analysis are reported in *Supplementary file 1*. For each experiment, both male and female mice were used and littermates were used when possible. All analyses were performed by 1 or more blinded investigators.

## Primary cultures and live imaging

Primary cortical cultures were derived from E14.5 embryonic dorsal cortices, as described (*Mitchell-Dick et al., 2019*), but with minor modifications: (1) cortices were trypsinized for 6 min and (2) 150,000 cells were plated on poly-D-lysine-coated glass-bottom 24-well culture plates (MatTek). Images were captured every 10 min and mitosis duration and cell division were identified as previously (*Pilaz et al., 2016*). Fate determination was performed post-imaging by immunostaining for Tuj1, Sox2, and Tbr2, as described (*Mitchell-Dick et al., 2019*).

## Plasmids, subcloning, and qRT-PCR analysis

*Ddx3y* sgRNAs were designed using Benchling and cloned into the pX330-U6-Chimeric_BB-CBh-hSpCas9 plasmid (AddGene # 42230) as described (http://www.addgene.org/crispr/zhang/). cDNA synthesis and qPCR were performed using the iScript Reverse Transcriptase and the iTaq Universal SYBR Green supermix (BioRad), respectively, per manufacturer's instructions. The primers for qRT-PCR all had an annealing temperature of 60 °C and relative expression was normalized to β-actin. See *Supplementary file 2* for sgRNAs and primers.

### Immunofluorescence

Embryonic brains were fixed and sectioned as previously (*Mao et al., 2015*). Coronal 20 µm sections from the somatosensory cortex were permeabilized with 1 X PBS/0.25% TritonX-100 and blocked with 5% NGS/PBS for 1 hr at room temperature. Sections were incubated with primary antibodies overnight at 4 °C, and secondary antibodies at room temperature for 1 hr (Alexa Fluor-conjugated, Thermo Fisher, 1:500). EdU staining was performed as previously (*Mitchell-Dick et al., 2019*). The following primary antibodies were used, rat: anti-SOX2 (Thermo Fisher, 14-9811-82, 1:1000), anti-BrdU (Abcam, ab6326, 1:200); mouse: anti-TUJ1 (Biolegend, 801202, 1:2000), anti-CTIP2 (Abcam, c8035, 1:500); rabbit: anti-TBR2 (Abcam, AB23345, 1:1000), anti-CC3 (Cell Signaling, 9661, 1:250), anti-NEUROD2 (Abcam, AB104430, 1:500), anti-Ki67 (Cell Signaling Technology, 12202, 1:1000), anti-PH3 (Millipore, 06–570, 1:500), anti-TBR1 (Cell Signaling Technology, 49,661 S, 1:1000), anti-Lhx2 (Millipore, ABE1402, 1:500), anti-DDX3X (Sigma Aldrich, HPA001648, 1:500); chicken anti-GFP (Abcam, Ab13970, 1:1000). Slides were mounted with Vectashield (Vector Labs, H-1000–10).

## Imaging and analysis

Images were captured using a Zeiss Axio Observer Z.1 equipped with an Apotome for optical sectioning at 10X, 20X, and/or 63X. For each experiment, 2–3 sections were imaged/embryo; images

were captured with identical exposures, cropped (200 or 300 μm radial columns), and brightness was equivalently adjusted across all images in Fiji. Cells were either manually (Fiji cell counter) or automatically (QuPath) counted. For QuPath, the following parameters were adjusted: requested pixel size = 0.1 μm, background radius = 5 μm, minimum area = 10 $μm^2$, maximum area = 200 $μm^2$, cell expansion = 2 μm, include cell nucleus and smooth boundaries were unchecked. The threshold was set for each individual channel, but equivalently across all sections (generally between 25 and 100). For binning analysis, 200 or 300 μm wide radial columns were divided into 5 or 10 evenly spaced bins spanning from the ventricular (bin 1) to the pial (bin 5 or 10) surface. Each cell was assigned to a bin to calculate the distribution.

### *In utero* electroporation

Plasmids were delivered to embryonic brains and IUEs were performed as previously (*Lennox et al., 2020*). Plasmids were used at the following concentrations: pCAG-GFP (1.0 μg/μL), pX330 empty or pX330-*Ddx3y* Ex2 sgRNA (2.4 μg/μL). Scrambled (Qiagen, 1022076) or *Setd3* siRNAs (Qiagen 1027416, Gene ID: 52690) were injected at 2.5 μM.

### EdU and BrdU injections

For cell cycle exit, EdU was administered by IP injection at 10 mg/kg to pregnant dams at E13.5 and embryos were harvested exactly 24 hr later. For semi-cumulative labeling, EdU was administered by IP injection at 10 mg/kg at t=0 followed by BrdU (30 mg/kg) at t=1.5 hr. The following calculations were used to derive Tc and Ts: S cells = BrdU + ; P cells = Ki67+; L cell fraction = EdU + BrdU- (ie: EdU +minus BrdU+); Ts=(S cells/L cells) * 1.5; Tc=(P cells/S cells) *Ts.

Single molecule fluorescence *in situ* hybridization (smFISH) smFISH probes against *Mus musculus Ddx3x* were designed and prepared as described (*Tsanov et al., 2016*). All solutions and buffers were prepared with diethyl pyrocarbonate, including PFA and sucrose, to quench RNAse activity. Twenty μm coronal sections were permeabilized in 0.5% Triton X-100 in PBS for 30 min at room temperature and rinsed twice with buffer containing 10% formamide and 2 x SCC buffer (Thermo, 15557044). smFISH probes were diluted 1:200 in buffer containing 10% formamide, 2 x SCC buffer and 10% dextran sulfate; 200 μL of diluted probes were added to each slide and incubated overnight at 37 °C. The next day, samples were washed two times at 37 °C and Hoechst was included in second wash. Slides were mounted with Vectashield (Vector Labs, H-1000–10). Slides were imaged at 63X. Puncta were counted in QuPath (*Bankhead et al., 2017*) in a 75 $μm^2$ box in the ventricular zone using a detection threshold of 0.2 and an expected detection size of 0.5 μm with the minimum being 0.4 and the maximum being 2.

## FACS

Samples were processed for FACS as described (*Mitchell-Dick et al., 2019*) and sorted at 6 °C using a B-C Astrios cell sorter with gates for forward scatter (FSC), side scatter (SSC), DAPI or Propidium Iodide (PI), GFP or TdTomato, and were sorted directly into RLT buffer and RNA was extracted (Qiagen RNAeasy kit; Qiagen, 74034).

### RNAseq and ribosome footprinting

Embryonic cortices were flash-frozen in liquid nitrogen and stored at –80 °C. For each n, four E11.5 microdissected cortices were thawed on ice and lysed in 400 μL of polysome buffer (20 mM Tris-HCl) pH 7.5, 150 mM NaCl, 5 mM MgCl2, 1 mM DTT, 100 μg/ml cycloheximide (CHX; Calbiochem, CAS 66-81-9), 25 U/ml Turbo DNase I (Invitrogen, AM2238) using a hand-blender. Lysates were pelleted at 2000 g for 10 min and titurated with a 26-gauge needle before adding TritonX-100% to 1%. Lysates were then clarified at 20,000 g for 10 min. A total of 100 μL of lysate was taken for RNA, to which 300 μL of RLT buffer was added and RNA was extracted using the RNAeasy kit (Qiagen, 74034). The remaining lysate was treated with 1 μL of RNase I (Ambion, AM2294) and incubated for 30 min at RT. After nuclease digestion, 1 μL of Superase In (Invitrogen, AM2694) was added, followed by 1 ml of TRIzol (Invitrogen, 15596026) and 200 μL of chloroform. The samples were vigorously shaken for 30 s and incubated for 3 min at RT before centrifugation at 12,000 g for 10 min at 4 °C. Approximately 750 μL of supernatant was mixed with 750 μL of isopropanol, 2 μL of glycoblue (Invitrogen AM9515), and 30 μL of sodium acetate (NaOAc) and frozen at –20 °C overnight. The next day, samples were

spun (20,000 g, 15 mins, 4 °C) and the pellet was washed 2 X with 80% ethanol and allowed to air-dry before resuspending the pellet in 7 µL of RNase-free water. A PNK reaction was then performed at 37 °C for 30 min on digested RNA by adding 1 µL of PNK buffer, 1 µL of 100 mM ATP, and 1 µL of PNK enzyme (NEB, M0247S). During the incubation, a 15% acrylamide denaturing urea-gel (Biorad, 4566053) was pre-run (200 V, 15 min, 1 X TBE buffer). Samples (containing 2 X sample dye Novex, LC6876) were run (~45 min, 200 V), along with dsDNA (Thermo, 10488023) and miRNA ladders (NEB, N2102S). Afterwards, the gel was incubated with SYBR gold (Invitrogen S11494) for 5 min and the RPFs (~25–30 nt) were isolated and frozen in 400 mM NaOAc at –80 °C. The following day, frozen gel fragments were thawed (95 °C, 5 min) and vortexed (20 min, 3 X). The supernatant was spun through a SpinX column (Corning, CLS8162) (max speed,10 mins) and 500 µL isopropanol and 2 µL glycoblue were added before freezing (–20 °C, 1 hr). The samples were centrifuged (max speed, 15 min, 4 °C) and washed 2 X with 80% EtOH. The pellet was resuspended in 5 µL of water and libraries were prepared using the Qiagen miRNA library prep kit (331502; followed protocol for 100 ng input and did 15 cycles of PCR amplification). Ribosomal RNA depletion was performed after the 3' adapter ligation step using the RiboMinus Eukaryote kit v2 (Thermo Fisher, A15020). For RNAseq cDNA library preparation, RNA concentrations were measured using a NanoDrop spectrophotometer (Thermo Scientific) and the quality of RNA was assessed using an Agilent fragment analyzer and the RNA Analysis Kit (15 nt; Agilent, DNF-471). cDNA libraries were generated using the Kapa mRNA HyperPrep kit with mRNA capture (KapaBiosystems, KR1352) using 50 ng input RNA.

## Sequencing and bioinformatic analysis

Three samples per condition per sex (WT and cKO) were sequenced, but one WT F had to be removed due to inefficient library prep. RNAseq and Ribo-seq libraries were sequenced on the NovaSeq 6,000 S-Prime with 150 bp paired-end reads and 75 bp single end reads, respectively. For RNAseq, libraries were sequenced to a depth of ~40–60 million total reads per sample. For Ribo-seq, libraries were sequenced to a depth of 50–70 million reads and only reads that uniquely mapped were retained. After adapter removal, reads were mapped to rRNA (*Langmead and Salzberg, 2012*) using bowtie2 v2.4.4, the remaining reads were mapped to GENCODE22 using STAR v2.7.9a (*Dobin et al., 2013*). Quality control of mapped reads and count matrices, using uniquely mapping reads only, were obtained using RibosomeProfilingQC (*Jianghong Ou, 2021*) and Ribo-seQC (*Calviello et al., 2019*). Changes in TE were calculated using DESeq2 by using assay type (RNA-seq or Ribo-seq) as an additional covariate. Translationally regulated genes were defined using an FDR <0.05 from a likelihood ratio test, using a reduced model without the assay type covariate, for example assuming no difference between RNA-seq and Ribo-seq counts (*Calviello et al., 2021*). ORFquant (*Calviello et al., 2020*) v1.02 was used to *de novo* identify translated ORFs using the pooled Ribo-seq data and Gencode M22 annotation as reference. Inputs to ORFquant were obtained using RiboseQC (*Calviello et al., 2019*). Only ATG-starting ORFs were detected using uniquely mapping reads only.

## Polysome profiling

Embryonic cortices were dissected and lysed identically to those used for Ribo-seq. Cortices from 2 embryos were pooled per n. Following clarification at 20,000 g, 50 µL of lysate was taken for RNA input. Clarified lysates were added to prepared 15–50% sucrose gradients (in the lysis buffer above, except lacking CHX and DNase) and ultracentrifuged (35,000 X g, 2 or 3.5 hr, 4 °C) using a SW41 Ti rotor. Following ultracentrifugation, 12 fractions (~1 ml each) were collected from each sample using a BioComp Piston Gradient Fractionator instrument fitted with a TRIAX flow cell to measure absorbance. RNA was extracted from 300 µL of each fraction or pooled fractions using TRIzol LS reagent (Thermo Fisher, 10296010). cDNA was synthesized from 250 ng of RNA using the iScript cDNA synthesis kit (Bio-Rad) (40 min, 46 °C), followed by RT-qPCR using gene-specific primers. Gene expression across different fractions was normalized to *B-actin* and then set relative to the corresponding expression from input.

## RNA immunoprecipitation (RIP)

Neuro2A cells were seeded into a six-well plate and transfected at confluency with Lipofectamine 2000 (ThermoFisher, 11668019) alone or with pCAG-GFP-human DDX3X per manufacturer protocol; after 4 hr, transfection media was exchanged for fresh media. At 24 hr post-transfection, cells were

subjected to RNA immunoprecipitation as previously (*Keene et al., 2006*) with modifications as listed here. Cells were scraped from the plate surface, washed twice in cold PBS, and lysed via trituration with a p200 pipette in 100 µL of lysis buffer (10 mM HEPES pH 7, 100 mM KCl, 5 mM MgCl2, 1 mM DTT, 0.5% NP-40). Following a 5-min centrifugation at 6500 g at 4 °C, 50 µL of supernatant was transferred to each of two tubes for immunoprecipitation, with the remaining ~10 µl of supernatant kept as input. A total of 193 µl of ice-cold NT2 buffer (50 mM Tris-HCl pH 7.4, 150 mM NaCl, 1 mM MgCl2, 0.05% NP-40) supplemented with 1 µL RNaseOUT (Invitrogen, 10777019) was added to the immunoprecipitation tubes together with 3.4 µg of anti-GFP (Santa Cruz, sc9996). This mix was incubated overnight at 4 °C with rotation. Fifty µl of magnetic beads solution (Protein G-coated Dynabeads, ThermoFisher Scientific, 0003D) were washed with 1 mL NT2 for each immunoprecipitation condition. Lysates were subsequently added to the washed beads and incubated for 4 hr at 4 °C with rotation. Beads were then washed five times with 1 mL NT2 buffer per wash. To prepare RNA for PCR analysis, 1 mL of TRIzol (ThermoFisher Scientific) was added to the beads and RNA was purified per the manufacturer's instructions. cDNA was prepared from 250 ng of RNA per condition using the iScript cDNA synthesis kit (Bio-Rad) according to the manufacturer's instructions. Resulting cDNAs were diluted 1:5, and 1 µL of this dilution was used as a PCR template for amplification using GoTaq Green Master Mix (Promega). Products were run on a 2% agarose gel and imaged using a BioRad GelDoc system.

## Cell lines

Neuro2A cells were obtained from ATCC, and their identity was authenticated based on imaging for morphology and qPCR analyses of mouse genes. Freshly purchased vials are used for these experiments. These cells were only used for transfection of siRNAs or constructs and confirmation of knockdown and expression. No cellular phenotypic assays were performed with these cells.

## SDS-PAGE and western blot analysis

The dorsal telencephalon of cortices from E14.5 mice were micro-dissected and flash frozen in liquid nitrogen and stored at –80 °C. Samples were lysed in 1 X RIPA buffer (Pierce, 89900) supplemented with protease inhibitors (Sigma, 78429) and titurated with a p1000 pipette and then spun at 15,000 g for 5 min at +4 °C. A BCA protein quantification was performed (Thermo, 23227) and 15 µg of protein was mixed with 2 X sample buffer (Biorad, 1610737) and heated at 95 °C for 10 min. Fifteen µL of sample was separated on a 12% polyacrylamide gel (Biorad, 4568046). The gel was transferred to a PVDF membrane (Biorad, 1704157) using the Trans-Blot Turbo system (Biorad). The membrane was then blocked in 5% milk/PBS-T for 1 hr at room temperature and then incubated overnight with anti-DDX3X (Sigma, HPA001648, 1:1000) and anti-B-actin (Santa Cruz, sc-47778, 1:500) in 5% milk/PBS-T at +4 °C. The membranes were washed 5 X with 1 X PBS-T and then incubated with anti-mouse and anti-rabbit HRP (Thermo, 32,430 and A16110, 1:2000) for 1 hr at room temperature in 5% milk/PBS-T. The membranes were again washed with 1 X PBS-T and then exposed using ECL (Thermo, 32106) and imaged on a ChemiDoc XRS+ (Biorad). Densitometry analysis was performed as previously described.

## Acknowledgements

We thank the *DDX3X* Foundation and members of the Silver, Floor, and Sherr labs for helpful discussions. This work was supported by the Holland-Trice Foundation, the DDX3X Foundation (DLS), NIH R21ND104514 (DLS), NIH R01NS120667 (DLS, SNF); Regeneration Next Initiative Postdoctoral Fellowship, NIH F32NS112566 (MLH); National Institutes of Health DP2GM132932 (SNF). S.N.F. is a Pew Scholar in the Biomedical Sciences, supported by The Pew Charitable Trusts. We thank Chris Nicchitta for help with Ribo-seq and Stacy Horner for use of the polysome profiling equipment. We thank Klaus-Nave (*Neurod6*-Cre/*Nex*-Cre) and Li-Ru You (*Ddx3x*^lox/lox) for mice. We thank the Duke Flow Cytometry and Genomics cores.

## Additional information

### Funding

| Funder | Grant reference number | Author |
| --- | --- | --- |
| National Institute of Neurological Disorders and Stroke | R21HD104514 | Debra L Silver |
| National Institute of Neurological Disorders and Stroke | R01NS120667 | Stephen N Floor Debra L Silver |
| National Institute of Neurological Disorders and Stroke | F32NS112566 | Mariah L Hoye |
| National Institute of General Medical Sciences | DP2GM132932 | Stephen N Floor |
| Pew Charitable Trusts | | Stephen N Floor |
| Holland-Trice Foundation | | Debra L Silver |
| DDX3X Foundation | | Stephen N Floor Debra L Silver |
| Regeneration Next Initiative | | Mariah L Hoye |

The funders had no role in study design, data collection and interpretation, or the decision to submit the work for publication.

### Author contributions

Mariah L Hoye, Conceptualization, Formal analysis, Funding acquisition, Investigation, Methodology, Validation, Writing – original draft, Writing – review and editing; Lorenzo Calviello, Formal analysis, Investigation, Software, Writing – review and editing; Abigail J Poff, Nna-Emeka Ejimogu, Carly R Newman, Formal analysis, Investigation, Validation, Writing – review and editing; Maya D Montgomery, Formal analysis, Investigation, Writing – review and editing; Jianhong Ou, Formal analysis, Software, Validation, Writing – review and editing; Stephen N Floor, Funding acquisition, Methodology, Supervision, Writing – review and editing; Debra L Silver, Conceptualization, Funding acquisition, Project administration, Resources, Supervision, Writing – original draft, Writing – review and editing

### Author ORCIDs

Mariah L Hoye ⓘ http://orcid.org/0000-0002-3430-2782
Abigail J Poff ⓘ http://orcid.org/0000-0001-6267-4201
Stephen N Floor ⓘ http://orcid.org/0000-0002-9965-9694
Debra L Silver ⓘ http://orcid.org/0000-0001-9189-844X

### Ethics

All animal use was approved by the Duke Institutional Animal Care and Use Committee (IACUC # A060-22-03) through 4/1/25. A statement was included in the manuscript.

### Decision letter and Author response

Decision letter https://doi.org/10.7554/eLife.78203.sa1
Author response https://doi.org/10.7554/eLife.78203.sa2

## Additional files

### Supplementary files

• Supplementary file 1. An excel file containing the comparison, sample size, statistical test and number of litters used (when applicable) for each data plot.

• Supplementary file 2. An excel file containing the primer and sgRNA sequences used in this paper.

• Supplementary file 3. An excel file containing three tabs (1) the differential expression analysis of

controls versus Ddx3x cKO for RNAseq and Riboseq data (using only unique reads); (2) the analysis of translational efficiency from wildtype (ie: control) data (using all reads); (3) the cell type-enriched transcripts used in *Figure 6B*.

• Transparent reporting form

### Data availability

Sequencing data has been deposited at GEO with the following accession number: GSE203078 . Supplemental excels for the sequencing data have been provided (Supplemental Excel 3).

The following dataset was generated:

| Author(s) | Year | Dataset title | Dataset URL | Database and Identifier |
|---|---|---|---|---|
| Hoye ML, Calviello L, Poff AJ, Ejimogu NE, Newman CR, Montgomery MD, Ou J, Floor SN, Silver DL | 2022 | Data from: Aberrant cortical development is driven by impaired cell cycle and translational control in a DDX3X syndrome model | https://www.ncbi.nlm.nih.gov/geo/query/acc.cgi?acc=GSE203078 | NCBI Gene Expression Omnibus, GSE203078 |

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
