## [Editor Report]

The paper beautifully documents the cortical developmental defects associated with DDX3X loss, detailing both the morphological, transcriptional, cell cycle, and protein translational defects in visually striking detail.

---

## [Decision Letter]

**Decision letter after peer review:**

Thank you for submitting your article "Aberrant cortical development is driven by impaired cell cycle and translational control in a DDX3X syndrome model" for consideration by *eLife*. Your article has been reviewed by 3 peer reviewers, and the evaluation has been overseen by a Reviewing Editor and Marianne Bronner as the Senior Editor. The reviewers have opted to remain anonymous.

Essential revisions:

1. It is unclear why the expression levels in the male cKO are reduced to levels comparable to the cHET females at D11.5 (Figure 1D, E), and not comparable to cKO females as one might expect. Could it be that the probes/primers used here are indeed specific for DDX3X or might they also detect DDX3Y. Is there is another explanation?

2. For their ribosomal profiling experiments, the authors focused on cKO females and males, while in the rest of the paper they argue that cKO males are actually comparable to cHET females. And then for polysome fractionation, they go back to cHET females. Those inconsistencies are not well justified in the manuscript. Since samples from cKO males and cKO females were used for the Ribo-seq and RNA-seq experiments, the rationale for using samples from cKO males and cHet females for the polysome fractionation experiment is unclear. Differences in RNA abundance attributed to developmental timepoints in RIBO seq vs. polysome fractionation (E11.5 vs. E14.5) might be due to different DDX3X levels.

3. Figure 6D used adjusted p-value <0.1 as the threshold, which is not a typical practice. We recommend filtering target genes with a more stringent standard.

4. The validity of the cKO mouse should be further confirmed with quantification of DDX3X protein levels, in addition to the mRNA analyses.

*Reviewer #1 (Recommendations for the authors):*

1. The authors should define the expression pattern of the Nex-Cre mouse in the text.

2. The authors should define different cell stainings better (e.g. what type of cells do Tbr-*Sox2*- cells represent).

3. How do the authors explain that there seem to be significantly more *Sox2*^+^/EdU+ cells in Figure 4D, but not *Sox2*^+^ Tbr2- EdU+ cells in Figure 4 – figure 3B?

4. In Figure 5D, what does the white bar represent? Only 1 division or no division at all? This should be clarified in the legend.

5. The reference RK, C.Y. et al., is not written correctly, the first author should be Yuen, R.K.C.

6. The authors comment on the spectrum of nonsense mutations that may be either LOF or hypomorphic. It is unclear to me what the authors mean by this, could they clarify this statement?

*Reviewer #3 (Recommendations for the authors):*

1) De novo DDX3X human mutations are associated with intellectual disability and over 80% of patients show abnormal corpus callosum. Do heterozygous Ddx3x cKO female mice and/or cKO males display comparable corpus callosum and learning/behavior phenotypes?

2) The authors suggested that Ddx3y loss of function phenocopies Ddx3x lof with the CRISPR IUE assay. This point could be further strengthened with another control – Ddx3y sgRNA IUE in female embryos, which probably had been collected in parallel with the samples shown in Figure 2D. In addition, given that ~15% of cells underwent apoptosis in homozygous cKO females, does Ddx3y sgRNA IUE in cKO males induce similar effects?

3) The authors described Figure 4G as E14.5 but the brain sections appear much thinner (than Figure 4A for example) and more proliferative than typical E14.5 tissues.

4) Figure 6D used adjusted p-value <0.1 as the threshold, which is not a typical practice. We recommend filtering target genes with a more stringent standard.

5) For Ddx3x dependent gene translation, are these Ribo-Seq targets directly regulated by Ddx3x? Are they overall enriched in certain biological functions?

---

## [Author Response]

Essential revisions:1. It is unclear why the expression levels in the male cKO are reduced to levels comparable to the cHET females at D11.5 (Figure 1D, E), and not comparable to cKO females as one might expect. Could it be that the probes/primers used here are indeed specific for DDX3X or might they also detect DDX3Y. Is there is another explanation?

We apologize for the lack of clarity regarding this point. We quantified that WT females had ~25% higher levels of *Ddx3x* mRNA expression than WT males (Figure 1G). Thus, we originally plotted males and females separately to illustrate the reduction of *Ddx3x* in conditional mice relative to their sex-matched controls (original Figures 1D, E). Because females have higher levels than males at baseline, the relative reduction in *Ddx3x* levels in cHet females and cKO males is similar, particularly at E11. In this revision, we now plot all of the sexes together which makes it easier to compare levels across all genotypes (new Figures 1E and 1F). Our probes are specific for *Ddx3x*, as evidenced in Figure 2C (in which we knockdown *Ddx3y* but observe no change in *Ddx3x*).

Importantly, males also express *Ddx3y*, which acts redundantly with *Ddx3x*. While there are no available DDX3Y specific antibodies, *Ddx3y* is upregulated at the RNA level in cKO males (Figure 2A). Thus we posit that the overall levels of DDX3 protein in males and females is relatively similar.

2. For their ribosomal profiling experiments, the authors focused on cKO females and males, while in the rest of the paper they argue that cKO males are actually comparable to cHET females. And then for polysome fractionation, they go back to cHET females. Those inconsistencies are not well justified in the manuscript. Since samples from cKO males and cKO females were used for the Ribo-seq and RNA-seq experiments, the rationale for using samples from cKO males and cHet females for the polysome fractionation experiment is unclear. Differences in RNA abundance attributed to developmental timepoints in RIBO seq vs. polysome fractionation (E11.5 vs. E14.5) might be due to different DDX3X levels.

We thank the reviewers for this suggestion. While we did use cKO females and males for the ribosome profiling experiment, this was done for several reasons.

1) Ribo-seq is more technically challenging than a standard RNAseq experiment, so we aimed to maximize the effect of *Ddx3x* knockout by using the cKO females. Because of the profound apoptosis that begins at E12.5 in cKO females, we opted to do these experiments at E11.5 to avoid potential complications due to cell death and composition changes.

2) The bulk of our paper focuses on the cortical development phenotypes of the cHet females and cKO males (to best model *DDX3X* syndrome), with significant phenotypes at E14.5. Thus, we initially performed the polysome fractionation of these genotypes at E14.5 to determine whether any of the DDX3X-dependent translation changes might be contributing to phenotypes at this stage. We did not include cKO females in this assay because at E14.5, most of the cells in the cortex are apoptotic.

In response to reviewer concerns, in this revised manuscript we include new polysome fractionation analyses at E11.5 using cKO females and cKO males-this provides validation of Ribo-seq of the same genotypes. These data show significant enrichment in monosome fractions for 2 targets (*Rcor2* and *Topbp1*) and trends for a 3^rd^ (*Setd3*, p=0.10). This also validates the same transcripts which are altered in polysomes at E14.5 in cHet females and cKO males. We include these new data in Figures 6H, J and Figure 6—figure supplement 2A,B.

We agree that differences in RNA abundance could be due to different developmental timepoints and DDX3X levels. We have included this important possibility in the discussion and removed this point from the results.

3. Figure 6D used adjusted p-value <0.1 as the threshold, which is not a typical practice. We recommend filtering target genes with a more stringent standard.

We originally used a threshold of adjusted p-value <0.1 because the Ribo-seq data is much more inherently noisy than a typical RNAseq dataset. However, to fit with more stringent standard practice we have re-graphed these data including only targets which are <0.05. These modified data are found in Figures 6D, E.

4. The validity of the cKO mouse should be further confirmed with quantification of DDX3X protein levels, in addition to the mRNA analyses.

We thank the reviewers for this suggestion. We have performed western blots using E14.5 brains from cHet females and cKO males (n=2-4 for each genotype and sex). cKO females were not used at E14.5 due to massive apoptosis. We quantified one western but repeated westerns several times. These data show: (a) WT females have higher levels than WT males (consistent with RNA levels); (b) cHet females have lower levels than WT females; c) cKO males have markedly reduced DDX3X, relative to all genotypes. These data corroborate qPCR analyses in Figure 1, and further validate the mouse model. In addition, the marked absence of DDX3X in males further confirms this antibody is specific to DDX3X and appears to not cross react with DDX3Y (which we note in the figure legend for these data). Please see Figure 1—figure supplement 1A. It is also valuable to note that the cKO mouse we are using has previously been validated by western analysis, in the initial study reporting this mouse [1]. Further, there are no available commercial antibodies specific for DDX3Y. As noted in response 1, we posit that the overall levels of DDX3 protein in males and females is relatively similar given the function of *Ddx3y*.

Reviewer #1 (Recommendations for the authors):1. The authors should define the expression pattern of the Nex-Cre mouse in the text.

We apologize for this oversight and have updated the text accordingly (p. 7).

2. The authors should define different cell stainings better (e.g. what type of cells do Tbr-Sox2- cells represent).

We have added additional descriptors to clarify this in the figure legends and in the results.

3. How do the authors explain that there seem to be significantly more Sox2^+^/EdU+ cells in Figure 4D, but not Sox2^+^ Tbr2- EdU+ cells in Figure 4 – figure 3B?

We specifically counted EdU+*Sox2*^+^Tbr2- and EdU+*Sox2*^+^Tbr2+ cells to try to tease apart possible differences in cell cycle exit between RGCs and immature IPs (*Sox2*^+^Tbr2+) cells. Although we didn’t see a significant increase in EdU+*Sox2*^+^Tbr2- subtypes, when we combined all *Sox2*^+^ cells (*Sox2*^+^Tbr2- and *Sox2*^+^Tbr2+), we saw a significant increase. However, we appreciate including all of these different sub-types is confusing, especially as these trends were not significant. In order to be consistent with the quantifications of progenitors in Figure 3, we have removed the analyses of these subtypes (old Figure 4—figure supplement 1B,C).

4. In Figure 5D, what does the white bar represent? Only 1 division or no division at all? This should be clarified in the legend.

We apologize for this oversight and have updated the figure and figure legend to clarify. The white bar represents cells that only divided once. We did not analyze non-dividing cells.

5. The reference RK, C.Y. et al., is not written correctly, the first author should be Yuen, R.K.C.

We have fixed this reference.

6. The authors comment on the spectrum of nonsense mutations that may be either LOF or hypomorphic. It is unclear to me what the authors mean by this, could they clarify this statement?

We meant to suggest that there may be a spectrum of mutations annotated as nonsense: those that act as a pure LoF (ie: an N-terminal nonsense mutation that undergoes NMD) versus a mutation that is hypomorphic (ie: a C-terminal mutation that escapes NMD and might have some functional protein activity). We have modified the wording on these to clarify (p. 18).

Reviewer #3 (Recommendations for the authors):Major questions:1) de novo DDX3X human mutations are associated with intellectual disability and over 80% of patients show abnormal corpus callosum. Do heterozygous Ddx3x cKO female mice and/or cKO males display comparable corpus callosum and learning/behavior phenotypes?

This is a fascinating question. We examined the corpus callosum using CUBIC along with LSL-Tomato reporters driven by *Emx1*-Cre in E18.5 in cHet females and cKO males (please see Author response image 1). We did not observe a striking defect. However, *DDX3X* patients have thin posterior corpus callosum and this likely necessitates analysis at later stages. Moreover, there may be non-cell autonomous contributions (outside the *Emx1* lineage). Thus, we feel this analysis is best done in future studies at later stages and using a germline model such as *Sox2*-Cre. Of note, a germline mouse model of *Ddx3x* cHET did not specifically describe callosal defects [2]. However, this study did report extensive behavioral defects of females. Of further relevance, the callosal defects in patients are most severe in individuals harboring missense mutations, so measuring the corpus callosum in missense models will be especially fruitful.

**Author response image 1. sa2fig1:** Representative 100um optical sagittal sections (50um on either side of the midsagittal plane) of brains of noted genotype generated using Imaris software. Scale bar = 300mm.

2) The authors suggested that Ddx3y loss of function phenocopies Ddx3x lof with the CRISPR IUE assay. This point could be further strengthened with another control – Ddx3y sgRNA IUE in female embryos, which probably had been collected in parallel with the samples shown in Figure 2D. In addition, given that ~15% of cells underwent apoptosis in homozygous cKO females, does Ddx3y sgRNA IUE in cKO males induce similar effects?

For these experiments we focused on collection and analysis of males, after confirming quantitatively that the gRNAs were specific to *Ddx3y* (Figure 2C). Please see Author response image 2 for some images of female embryos treated with *Ddx3y* sgRNAs. Our preliminary analysis of two IUE’d brains suggests there is no striking phenotype. Because our n’s are low, we did not quantify this.

**Author response image 2. sa2fig2:** Representative images of female cortices electroporated with *Ddx3y* gRNA.

We tried the suggested experiment, to IUE *Ddx3y* sgRNAs into cKO males, but encountered technical difficulties with obtaining good IUEs, perhaps because of delayed cell cycle, and other neurogenesis defects already present in this genotype.

3) The authors described Figure 4G as E14.5 but the brain sections appear much thinner (than Figure 4A for example) and more proliferative than typical E14.5 tissues.

Figures 4A and 4G were brains from separate experiments and litters. For experiments in 4G we evaluated sections which were slightly more medial, which explains why these look “younger”. We confirmed these are indeed E14.5. We have clarified this in the figure legend. In previous cell cycle analyses by our lab at E13.5 the Tc is much lower, at15 hours. While these sections were more medial, our live imaging of the whole dorsal telencephalon supports the conclusion that cell cycle is longer in Ddx3x deficient progenitors (Figure 5C).

4) Figure 6D used adjusted p-value <0.1 as the threshold, which is not a typical practice. We recommend filtering target genes with a more stringent standard.

Please see our response #3 to Essential Revisions.

5) For Ddx3x dependent gene translation, are these Ribo-Seq targets directly regulated by Ddx3x? Are they overall enriched in certain biological functions?

This is a fascinating question and thank you for the suggestion. We have thus performed DDX3X RNA immunoprecipitation for the same candidates we validated in Figure 6 (*Setd3*, *Rcor2*, and *Topbp1*). These new data show these 3 key translational targets are bona fide targets of DDX3X. These data are now included in Figure 6—figure supplement 2D. With so few genes we did not see a strong bias in GO analysis, only a few pathways are weakly enriched (nucleus, RNA binding) and thus, we did not include this analysis. However, we do note in the discussion that several targets are epigenetic regulators.

References

1. Chen, C.-Y., et al., *Targeted inactivation of murine Ddx3x: essential roles of Ddx3xin placentation and embryogenesis.* Human molecular genetics, 2016: p. ddw143-18.

2. Boitnott, A., et al., *Developmental and Behavioral Phenotypes in a Mouse Model of DDX3X Syndrome.* Biol Psychiatry, 2021. 90(11): p. 742-755.